# Deficiency of CAMSAP2 impairs olfaction and the morphogenesis of mitral cells

Zhengrong Zhou [ID] [1,2,8 ✉], Xiaojuan Yang[1,3,8], Aihua Mao [ID] [4], Honglin Xu[1], Chunnuan Lin[1,3], Mengge Yang[1,3], Weichang Hu[1,3], Jinhui Shao[1,3], Peipei Xu[1,3], Yuejia Li[1,3], Wenguang Li[5], Ruifan Lin[6], Rui Zhang[1], Qi Xie[6], Zhiheng Xu [ID] [1,3,7] & Wenxiang Meng [ID] [1,3,7 ✉]

## Abstract

In developing olfactory bulb (OB), mitral cells (MCs) remodel their dendrites to establish the precise olfactory circuit, and these circuits are critical for individuals to sense odors and elicit behaviors for survival. However, how microtubules (MTs) participate in the process of dendritic remodeling remains elusive. Here, we reveal that calmodulin-regulated spectrin-associated proteins (CAMSAPs), a family of proteins that bind to the minus-end of the noncentrosomal MTs, play a crucial part in the development of MC dendrites. We observed that *Camsap2* knockout (KO) males are infertile while the reproductive tract is normal. Further study showed that the infertility was due to the severe defects of mating behavior in male mice. Besides, mice with loss-of-function displayed defects in the sense of smell. Furthermore, we found that the deficiency of CAMSAP2 impairs the classical morphology of MCs, and the CAMSAP2-dependent dendritic remodeling process is responsible for this defect. Thus, our findings demonstrate that CAMSAP2 plays a vital role in regulating the development of MCs.

**Keywords** CAMSAP2; Mitral Cell; Odor-dependent Behaviors; Morphogenesis
**Subject Categories** Cell Adhesion, Polarity & Cytoskeleton; Development; Neuroscience

## Introduction

In the mammalian olfactory system, the odorant receptors (ORs) in the nasal cavity recognize chemical cues and relay the information to the OB. After processing in the OB, mitral cells and tufted cells (MTCs), the only two projection neurons in OB, convey the odor information to the olfactory cortex to induce distinct behaviors (Chen and Hong, 2018; Dulac and Wagner, 2006). Olfactory sensory neurons (OSNs) expressing the same OR converge and send their axons to two specific sites of the OB to form a glomerular structure (Mombaerts et al, 1996). In the OB, MTCs extend a single apical dendrite radially into the glomerulus to receive the odor information (Dulac and Wagner, 2006; Imai, 2014). As OSN axons in one glomerulus represent one OR, and each MTC projects its primary dendrite to only one glomerulus, this junction constructs the precise connection between ORs and olfactory cortex (Imamura et al, 2011). However, how neurons shape the precise connection between OSNs and MTCs remains elusive.

During embryonic development, the MCs migrate from the olfactory ventricle to the mitral cell layer (MCL), and extend dendrites into multiple glomeruli (Blanchart et al, 2006; Hinds, 1968). After the process of pruning in the first postnatal week, these projection neurons form the typical mature morphology to connect with OSNs and interact with local interneurons to construct olfactory circuits (Imai, 2014; Lledo et al, 2005). The odorous signal input from the glomerulus is first processed by local circuits before it is relayed by projection neurons to the olfactory cortex. Thus, the precise morphogenesis of projection neurons, that is, one apical dendrite extended into the glomerulus and several lateral dendrites extended into the external plexiform layer (EPL), is critical for odor information processing and subsequent behaviors. Abolishment of odorant-evoked activity only delays but does not perturb the pruning of MCs (Lin et al, 2000; Matsutani and Yamamoto, 2000), and genetic manipulation alters the organization of MCs (Inokuchi et al, 2017; Kobayakawa et al, 2007). Moreover, the BMP and Notch pathways were reported to regulate the morphogenesis of MCs dendrites (Aihara et al, 2021; Muroyama et al, 2016). Besides, activity-dependent local protection participates in this remodeling process, mainly via regulating the formation of F-actin (Aihara et al, 2021; Fujimoto et al, 2023). However, whether and how MTs regulate the remodeling of MCs remains unclear.

The CAMSAPs are MT minus-ends binding proteins and play a vital role in regulating the MT network (Akhmanova and Hoogenraad, 2015; Meng et al, 2008; Tanaka et al, 2012; Zhou et al, 2020). Unlike *Drosophila* and *C. elegans*, mammals have three homologs, CAMSAP1, CAMSAP2 (CAMSAP1L1) and CAMSAP3

[1]State Key Laboratory of Molecular Developmental Biology, Institute of Genetics and Developmental Biology, Chinese Academy of Sciences, 100101 Beijing, China. [2]Neuroscience Center, Department of Basic Medical Sciences, Shantou University Medical College, 515041 Shantou, Guangdong, China. [3]University of Chinese Academy of Sciences, 100049 Beijing, China. [4]Biology Department, College of Sciences, Shantou University, 515063 Shantou, China. [5]Animal Laboratory Center, Institute of Genetics and Developmental Biology, Chinese Academy of Sciences, 100101 Beijing, China. [6]Chinese Academy of Chinese Medical Sciences, 100700 Beijing, China. [7]Innovation Academy for Seed Design, Chinese Academy of Sciences, 100101 Beijing, China. [8]These authors contributed equally: Zhengrong Zhou, Xiaojuan Yang. ✉E-mail: zrzhou@genetics.ac.cn; wxmeng@genetics.ac.cn

(Nezha) (Akhmanova and Hoogenraad, 2015). Though the structural domains of CAMSAP proteins are similar, they are non-redundant proteins as a series of discrepancies exist among them. First, their behavior is different. CAMSAP1 only tracks the minus-ends of MTs while CAMSAP2 and CAMSAP3 decorate and stabilize the minus-ends (Hendershott and Vale, 2014; Jiang et al, 2014; Tanaka et al, 2012); Second, they have different expressions, locations and distributions in cells and tissue (Ho et al, 2023; Pongrakhananon et al, 2018; Wu et al, 2016; Zhou et al, 2020). Finally, their functions are divergent. CAMSAP1 regulates neuronal migration and polarization, while CAMSAP3 is only responsible for neuronal polarization (Khalaf-Nazzal et al, 2022; Pongrakhananon et al, 2018; Zhou et al, 2020). Besides, the previous study and our recent work also show that CAMSAP1 but not CAMSAP2 and CAMSAP3 is responsible for spermatogenesis (Hu et al, 2023; Robinson et al, 2020). These features suggest that CAMSAPs are non-redundant in mammals and regulate mammals' development in a sophisticated manner. Previous studies showed that the deficiency of CAMSAP1 results in epileptic seizure and neuronal migration disorder (Khalaf-Nazzal et al, 2022; Zhou et al, 2020), while *Camsap3* knockout (KO) mice exhibit primary ciliary dyskinesia (Robinson et al, 2020). In this study, we set out to similarly address the function of CAMSAP2 and its potential role in organismal development.

Herein described, we construct *Camsap2* KO and conditional knockout (cKO) mice to investigate the function of CAMSAP2 in organismal development. We observed that a deficiency of CAMSAP2 abolished the mating behavior of male mice and led to infertility. *Camsap2* deletion perturbs the morphogenesis of MC dendrites and the sense of smell.

# Results and discussion

## *Camsap2* KO males are infertile, while spermatogenesis is normal

To investigate the function of CAMSAP2, we generated a line of *Camsap2* null allele mice via CRISPR/Cas9 (Appendix Fig. S1A). Animals carrying both null alleles are designated *Camsap2*(−/−) (or *Camsap2* KO) (Appendix Fig. S1A–C). In comparison with the slight body weight reduction in *Camsap2*(−/−), the most significant phenotype is the sterility of *Camsap2*(−/−) males, as we could not obtain pups with those males (Fig. 1A; Appendix Fig. S1D,E).

Considering CAMSAP2 is a noncentrosomal MT minus-ends binding protein, and the spermatozoa contain abundant MT structures, such as the manchette and the sperm flagella (Lehti and Sironen, 2016; O'Donnell, 2014; Rosenbaum and Witman, 2002), the most reasonable cause of male sterility in *Camsap2*(−/−) mice was the impairment of sperm formation. We therefore performed Hematoxylin and eosin (H&E) staining to analyze the structure of the testis and sperm, but found that both were normal in *Camsap2*(−/−) males (Fig. 1B), and the percentage of motile sperm from *Camsap2*(−/−) males were comparable to wildtype (WT) (Fig. 1C). To further confirm that the sperm from *Camsap2*(−/−) males was fully functional, we performed in vitro fertilization (IVF) assays and found that the percentage of 2-cell embryos was similar to the WT condition (Fig. 1D,E). Furthermore, sperm from *Camsap2*(+/+) or *Camsap2*(−/−) used in artificial

insemination led to roughly equal numbers of successfully born pups (Fig. 1F). Collectively, these results indicate that CAMSAP2 controls male reproduction independent of the reproductive tract.

## Ablation of CAMSAP2 alters the sexual behavior of male mice

Since the sperm from *Camsap2*(−/−) males can generate offspring via the artificial insemination, we suspected that the deficiency of CAMSAP2 may change the copulatory behavior of male mice. To test this assumption, male mice were paired with female mice and the presence of vaginal plugs (indicative of ejaculation) was assessed for consecutive 5 days. The production of plugs caused by *Camsap2*(+/+) males was detected at a mean of 3 days (Fig. 1G). However, we could not detect vaginal plugs in females paired with *Camsap2*(−/−) males (Fig. 1G), indicating that *Camsap2*(−/−) males did not successfully mate with females. In male mice, copulation involves approach and investigation, emission of ultrasonic vocalizations (USVs), mounting, intromission and ejaculation (Chen and Hong, 2018; Hull and Dominguez, 2007). To find the defective phase, we then analyze the sexual behavior of males with video recordings. Both *Camsap2*(+/+) and *Camsap2*(−/−) males showed no difference in duration and frequency of sniffing, while the mounting behavior was almost completely absent in *Camsap2*(−/−) males (Fig. 1H,I). Furthermore, we examined the USVs emitted by male mice in response to female mice, which is regarded as a male-typical precopulatory reproductive behavior normally accompanied by sexual chasing, preceding mounts or intromission. Besides, male who emits USVs often exhibit higher attractiveness to female and this process also enhances sexual receptivity in female (Asaba et al, 2014; Nyby et al, 1992; Pomerantz et al, 1983). During the experiment, most *Camsap2*(+/+) males vocalized intensively, whereas none of the *Camsap2*(−/−) males vocalized (Fig. 1J,K). To exclude the possibility that loss of CAMSAP2 destroys the ability to generally produce ultrasound, P6 mice pups were separated from their mother and both *Camsap2*(+/+) and *Camsap2*(−/−) mice emitted ultrasound as a separation response (Fig. EV1A,B). These results indicate the deficiency of CAMSAP2 impairs the activation of sexual motivation in male mice.

Steroid hormones activate the emission of USV and mounting in response to stimulus females. Moreover, the gonadal steroid hormones also play a critical role in the development and function of these behavior-associated neural circuits (Hull and Dominguez, 2007; Nyby et al, 1992; Wu et al, 2009), we then assayed the titers of hormones in serum. Unexpectedly, the titers of circulating testosterone and estradiol in *Camsap2*(−/−) males were comparable to those of *Camsap2*(+/+) males (Fig. EV1C,D). As motor deficit or muscular weakness may limit the mounting behavior, we also assayed their motor performance on the rotarod, and *Camsap2*(−/−) males performed equivalently to *Camsap2*(+/+) males (Fig. EV1E). Taken together, these results suggest that CAMSAP2 is required for sexual behaviors.

## Loss of CAMSAP2 impairs the sense of smell

As *Camsap2*(−/−) males sniff but do not mate with the receptive females (Fig. 1G–I), we then asked whether it is an olfaction defect that might cause these results. To test this hypothesis, *Camsap2*(+/+)

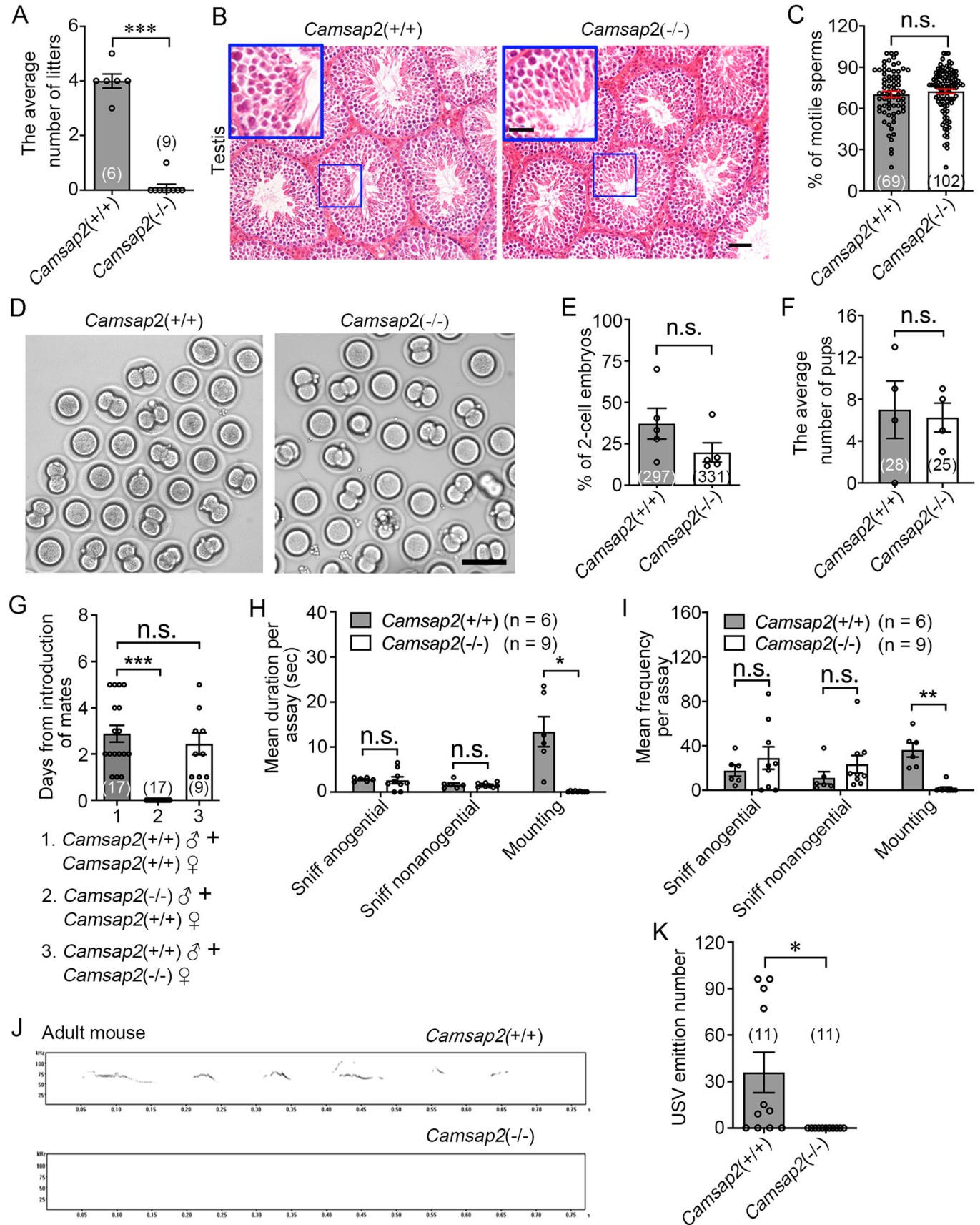

**Figure 1.  Copulatory behavior but not spermatogenesis is responsible for the infertility of *Camsap2* KO males.**

(A) Quantification of the average number of litters produced by male mice (unpaired Student's *t* test, $n = 6$ or 9 biological replicates, ***$P = 2.84 \times 10^{-6}$). (B) Representative images of testicular histology sections stained with H&E staining. (C) The percentage of motile sperms produced by *Camsap2*(+/+) and *Camsap2*(−/−) mice (unpaired Student's *t* test, $n = 4$ biological replicates). (D) Representative images of 2-cell embryos produced by IVF. (E) The percentage of 2-cell embryos produced by IVF (unpaired Student's *t* test, $n = 5$ biological replicates). (F) The average number of pups produced by IVF (unpaired Student's *t* test, $n = 4$ biological replicates). (G) Mice mating assay. Average number of days after pairing until the copulatory plug appears, 0. indicates that no plugs appeared after a maximum time span of 5 days of monitoring (one-way ANOVA with Dunnett's multiple comparisons test, $n = 9$–17 biological replicates, ***$P < 0.0001$). (H, I) The duration (H) and frequency (I) of mating subroutines in WT and KO mice (unpaired Student's *t* test, $n = 6$ or 9 biological replicates, *$P = 0.0102$, **$P = 0.00233$). (J) Representative spectrograms of USVs emitted by adult male mice. (K) Quantification of the number of USVs emitted by adult male mice (unpaired Student's *t* test, $n = 11$ biological replicates, *$P = 0.0208$). Data information: Data are represented as mean ± SEM. n.s. not significant, $P > 0.05$. Scale bars: (B) 100 µm and 50 µm in full-size images and zoomed areas; (D) 50 µm. Source data are available online for this figure.

males and *Camsap2*(−/−) males were placed in a cage that contained male and female urine. *Camsap2*(+/+) males discriminate between the two types of urine and show a higher preference for female urine, while *Camsap2*(−/−) males cannot distinguish between them (Fig. 2A). Subsequently, we examined the territorial marking behavior, a male-typical territorial behavior of depositing urine spots in response to the estrous female scent (Schoeller et al, 2016). We found that *Camsap2*(+/+) males robustly marked their territory, while *Camsap2*(−/−) males did not enhance their marking behavior when female urine was introduced into the cage (Fig. 2B).

As most mammalian species have two olfactory circuits, the accessory system to sense the non-volatile pheromonal odorants from conspecifics and the main olfactory system to detect non-pheromonal odors from food and predators which is relevant to species survival (Baum and Cherry, 2015), we then asked whether both olfactory circuits are impaired in *Camsap2*(−/−) mice. To answer this question, a buried food test was performed to test general olfactory function, and the latency for finding the hidden food was significantly longer in *Camsap2*(−/−) mice (Fig. 2C). Defensive behaviors, including avoidance and risk assessment, are elicited when mice sensed their predators' odors (Papes et al, 2010; Schoeller et al, 2016). We, therefore, placed individual mice in a three-chamber cage with a live (anesthetized) rat, which is considered a natural mouse predator. *Camsap2*(+/+) mice displayed significant innate risk assessment and avoidance behaviors and spent most of the test time in the hiding chamber. In contrast, the *Camsap2*(−/−) mice did not display these defensive behaviors, rather, they directly investigated the rat without apparent caution or retreating (Fig. 2D–F). Together, our results demonstrate that deficiency of CAMSAP2 causes a general olfactory defect, which leads to *Camsap2*(−/−) mice being unable to detect the odors of conspecifics, predators and food.

To confirm that the abnormal behaviors mentioned above are due to the defects of the nerve system, we generated a conditional allele with LoxP sites flanking *Camsap2* exon 2 (Appendix Fig. S2A,B). These mice were crossed with the Nesin-Cre line, a pan-neural line that expresses Cre-recombinase throughout the brain, and bred to obtain homozygous *Camsap2*fl/fl; Nestin-Cre (denoted *Camsap2*cKO) mice (Appendix Fig. S2C). *Camsap2*cKO mice show smaller brain and body sizes compared to *Camsap2*fl/fl mice (Appendix Fig. S2D-F). In correspondence with *Camsap2* null allele mice, *Camsap2*cKO mice are also infertile (Fig. 2G), and almost none of them produce vaginal plugs after cohabiting with female mice (Fig. 2H). In addition, *Camsap2*cKO mice did not display innate risk assessment and avoidance behaviors (Fig. 2I,J). Thus, by analyzing

the behaviors of mice, we have determined that the expression of CAMSAP2 in the nerve system is required for the sense of smell.

## CAMSAP2 is required for the development of MCs

As the OB is an important station for odor information processing and delivery. We were then curious whether the loss of CAMSAP2 affects the OB. As we expected, the structure of the OB was changed (Fig. EV2A,B), and especially the thickness of the EPL was significantly reduced in knockout mice (Fig. EV2A–E).

EPL is a neuropil layer that mainly contains the dendrites and synapses of MTC, MTC are the only two projection neurons in the OB that directly receive excitatory input from OSNs. After processing, they convey sensory information to the olfactory cortex (as MCs and TCs are similar in morphology and function, hereafter, we focus our research only on MCs). We then investigated whether the *Camsap2* gene disruption in MCs leads to variation in the anatomical properties of MCs. We first immunostained with anti-neurofilament 165 (NF165) to label the dendrites in the EPL. In control mice, MCs sent thick apical dendrites straight toward the GL (Fig. 3A). In contrast, the classical characteristic of apical dendrites disappeared in *Camsap2*cKO mice (Fig. 3A). Subsequently, we analyzed MCs and their dendrites by staining with anti-PGP9.5, a specific marker for MCs (Taniguchi et al, 1993). The apical dendrites were easily visualized in control mice, while they became indistinct in *Camsap2*cKO mice (Fig. 3B).

To investigate the morphological changes of single MCs, we stained the OB by Golgi-cox staining. The typical mature morphology of MCs was seen in *Camsap2*fl/fl mice. However, the proportion of MCs with normal morphology in *Camsap2*cKO was lower than that in *Camsap2*fl/fl (Fig. 3C,D), and we found some MCs in *Camsap2*cKO showing atypical morphology, such as multiple apical dendrites or the apical dendrite extending toward the lateral side (Fig. 3C). As Golgi-cox staining sporadically labels all neurons and their processes in the OB, this feature impeded us from acquiring comprehensive information about single MCs. To resolve this problem, we combined the AAV-mediated gene delivery system and the supernova system to specifically label MCs at single-cell resolution (Luo et al, 2016; Togashi et al, 2020). The virus mixture was unilaterally injected to the OB ventricle at E14.5, and the OB was collected at postnatal day (P) 21–23. After immunostaining and a tetrahydrofuran-based clearing procedure (Erturk et al, 2011), the morphology of MCs was analyzed (Fig. 3E). Corresponding to the results of Golgi-cox staining, mScarlet labeled MCs showed multiple morphologies in MC-specific knockout mice

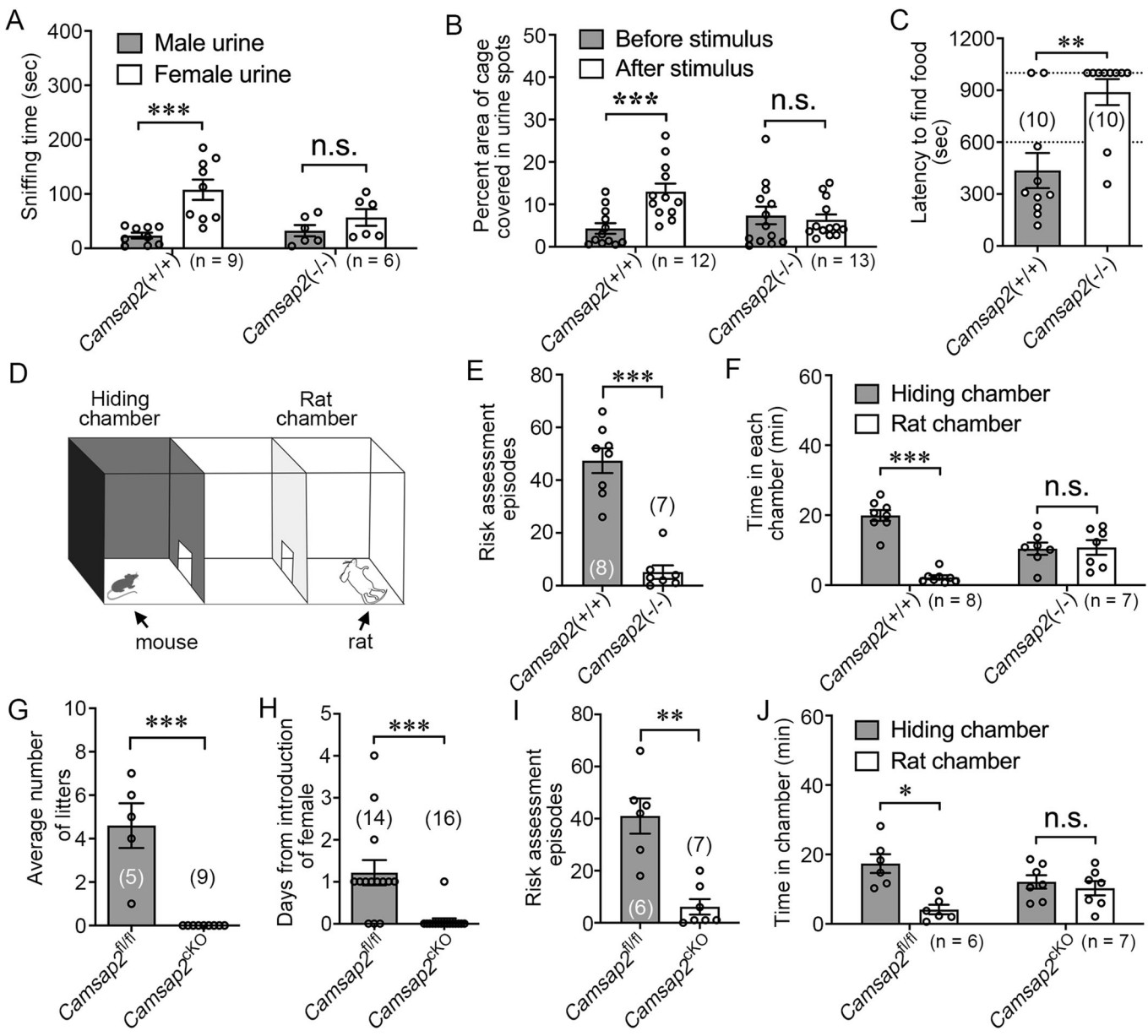

**Figure 2. Deficiency of CAMSAP2 dysregulated olfactory behaviors.**

(A) Total amount of time that male mice spend sniffing male or female urine (ratio paired Student's t test, n = 9 or 6 biological replicates, ***P = 0.0001). (B) Territorial marking assay. Percent area of cage covered by male urine in response to a spot of female urine (ratio paired Student's t test, n = 12 or 13 biological replicates, ***P = 0.0006). (C) Buried food test. The latency for mice to locate the food pellet. The dotted line at 600 indicates the end time of the experiment, and the dotted line at 1000 indicates the mice do not find the hidden food within the allotted time (unpaired Student's t test, n = 10 biological replicates, **P = 0.00233). (D) Schematic diagram of the defensive behavior test arena. The mouse and rat in each chamber are indicated by arrows. (E) Quantification of risk assessment behavior (unpaired Student's t test, n = 8 or 7 biological replicates, ***P = 8.95 × 10⁻⁶). (F) Quantification of the time spent by mice in the hiding chamber and rat chamber (ratio paired Student's t test, n = 8 or 7 biological replicates, ***P = 0.0001). (G) Quantification of the average number of litters produced by Camsap2^fl/fl and Camsap2^cKO male mice, 0 indicate that no plugs appeared after a maximum time span of 6 days of monitoring (unpaired Student's t test, n = 5 or 9 biological replicates, ***P = 0.0005). (H) Mice mating assay of Camsap2^fl/fl and Camsap2^cKO mice. The average number of days after pairing until the copulatory plug appears with a maximum time span of 6 days (unpaired Student's t test, n = 14 or 16 biological replicates, ***P < 0.0001). (I) Quantification of risk assessment behavior of Camsap2^fl/fl and Camsap2^cKO mice (unpaired Student's t test, n = 6 or 7 biological replicates, **P = 0.00226). (J) Quantification of the time spent by Camsap2^fl/fl and Camsap2^cKO male mice in the hiding chamber and rat chamber (ratio paired Student's t test, n = 6 or 7 biological replicates, *P = 0.026). Data information: Data are represented as mean ± SEM. n.s. not significant, P > 0.05. Source data are available online for this figure.

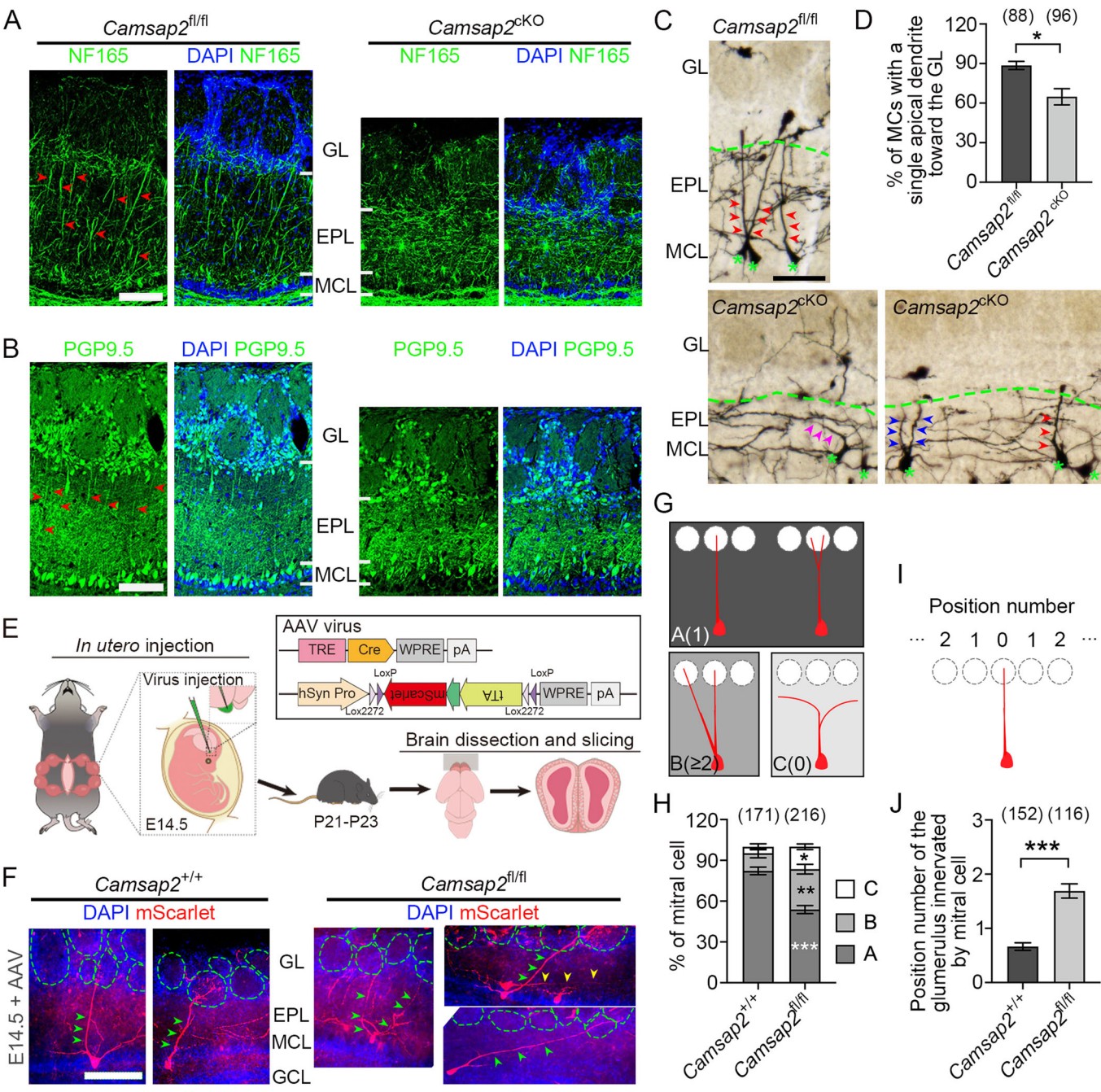

**Figure 3. CAMSAP2 deficiency disturbs the apical dendritic projection of MCs.**

(A, B) Representative images of immunofluorescence staining for NF165 (A) and PGP9.5 (B) in OB coronal sections. Red arrowheads indicate the perpendicular apical dendrites of MCs. (C) Neurons in the OB were stained by Golgi-cox staining. Green asterisks indicate the MCs, red arrowheads indicate the single apical dendrites extending from the MC straight toward the GL, blue arrowheads indicate two apical dendrites extending from the single MC, pink arrowheads indicate single apical dendrites extending towards the lateral side of EPL. Green dashed lines indicate the boundary of GL/EPL. (D) Quantification of the percentage of MCs with a perpendicular single apical dendrite (unpaired Student's t test, n = 5 biological replicates, *P = 0.0137). (E) Schema showing AAV-based sparse labeling. An AAV mixture was injected into the olfactory ventricle at E14.5, the neuronal morphology was analyzed at P21-P23. (F) Representative images of MCs labeled by mScarlet. Apical dendrites extending toward GL are indicated by green arrowheads, and an apical dendrite extending toward the lateral side is indicated by yellow arrowheads. Glomeruli are depicted by green dashed circles. (G) Apical dendrite morphologies of MCs were divided into 3 types: Type A, apical dendrite projecting into a single glomerulus; Type B, apical dendrite projecting into multiple glomeruli; Type C, apical dendrite does not project into any glomerulus. (H) Quantification of the percentage of MCs with different types of apical dendrite (two-way ANOVA with Šídák's multiple comparisons test, n = 6 or 5 biological replicates, *P = 0.0228, **P = 0.001, ***P < 0.0001, detailed information is reported in Source data). (I) Schema showing the position number of glomeruli innervated by an MC. (J) Quantification of the position number of glomeruli innervated by MCs (unpaired Student's t test, n = 6 or 5 biological replicates, ***P = 7.75 × 10⁻¹¹). Data information: Data are represented as mean ± SEM. Scale bars, 100 μm. Source data are available online for this figure.

(Fig. 3F), which we then categorized into three types (Fig. 3G). Compared with the MCs in WT mice, MCs in MC-specific knockout mice showed a higher proportion of type B and C (Fig. 3H). Furthermore, we also found that many more MCs in MC-specific knockout mice tend to project to a neighboring glomerulus (Fig. 3F,I,J), and select adjacent targets requiring longer dendrites (Position number 1 or 2, Fig. 3I). Even though we do not analyze the lateral dendrites of MCs, the narrowed EPL suggests that deficiency of CAMSAP2 impairs the development of lateral dendrites. Collectively, our data show that CAMSAP2 participates in the morphogenesis of MCs.

## CAMSAP2 deficiency alters dendritic remodeling

The development of MCs is composed of two main stages: neurogenesis, migration and neurite extension in the embryonic stage, and dendrite remodeling (pruning and stabilization) in the first postnatal week (Fig. 4A) (Blanchart et al, 2006; Hinds, 1968;

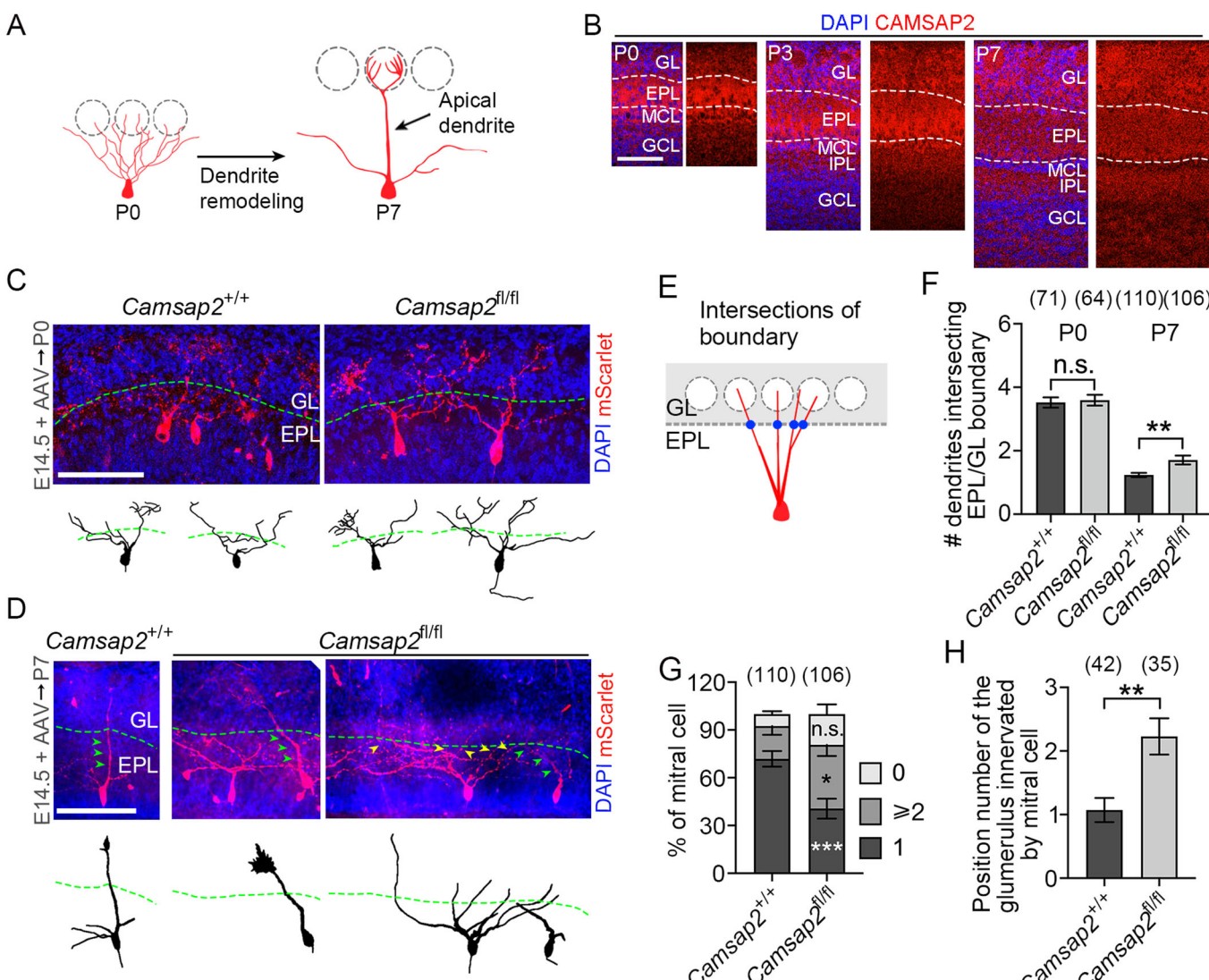

**Figure 4. CAMSAP2 deficiency disrupts dendritic remodeling.**

(A) Development of MC from P0 to P7. At P0, the MC develops widespread apical dendrites into multiple glomeruli, at P7, the MC forms typical mature morphology with a single apical dendrite and several lateral dendrites via dendritic remodeling. (B) Distribution of CAMSAP2 in coronal sections of the OB at P0, P3, and P7. (C, D) Representative images of MCs labeled by mScarlet. Green arrowheads indicate single apical dendrites extending from the MC toward the GL, and yellow arrowheads indicate multiple apical dendrites from single MC extending toward the GL. Green dashed lines indicate the boundary of GL/EPL. Representative MCs are reconstructed and shown below. (E) Intersections between dendrites and the GL/EPL boundary are shown as a cartoon. (F) Quantification of the number of crossing sites at the GL/EPL border in P0 and P7 mice (two-way ANOVA with Šídák's multiple comparisons test, P0, $n = 4$ or 6 biological replicates; P7, $n = 7$ or 6 biological replicates, $**P = 0.0098$, detailed information is reported in Source data). (G) Quantification of the percentage of MC with different number of intersections in P7 mice (two-way ANOVA with Šídák's multiple comparisons test, $n = 7$ or 6 biological replicates, $*P = 0.0444$, $***P = 0.0007$, detailed information is reported in Source data). (H) Quantification of the position number of glomeruli innervated by MC apical dendrites (unpaired Student's $t$ test, $n = 7$ or 6 biological replicates, $**P = 0.00127$). Data information: Data are represented as mean ± SEM. n.s. not significant, $P > 0.05$. Scale bars, 100 μm. Source data are available online for this figure.

Imai, 2014; Imamura et al, 2011; Ravi et al, 2017). To determine which developmental stage CAMSAP2 participates in, we first analyzed the proliferation and migration of MCs. The fraction of Pax6$^+$BrdU$^+$/Pax6$^+$ neuron was equivalent (Fig. EV3A,B), moreover, the density of MCs in the MCL was also unchanged (Fig. EV4B–E), suggesting that the neurogenesis of MCs is normal in CAMSAP2 deficient mice. We also excluded the possibility that deficiency of CAMSAP2 leads to the defects of apoptosis or migration in *Camsap2* KO or cKO mice (Figs. EV3C,D and EV4). However, the boundary of the MCL became obscure (Fig. EV4B,D). This result hinted that CAMSAP2 may participate in the process of dendrite remodeling, as the soma of MCs is aligned to form the MCL during this process (Ravi et al, 2017).

To verify our hypothesis, we first analyzed the distribution pattern of CAMSAP2 in the OB at the postnatal stage. CAMSAP2 showed high intensity in the EPL during remodeling, but this regional preference disappeared as the remodeling process finished in P7 (Fig. 4B). To further confirm that CAMSAP2 functions in regulating the remodeling of apical dendrites, the AAV virus was injected at E14.5 as before (Fig. 3E), while the morphology of MCs was analyzed at P0 and P7, the start and end of the pruning process. Since MCs at P0 do not form the typical mature morphology, we cannot divide the MCs into categories as before (Fig. 3G). For this reason, we counted the number of apical dendrites crossing the GL/EPL boundary (Fig. 4C,E), as these dendrites participate in the subsequent remodeling procedure. We found that the intersection numbers between CAMSAP2 deficient MCs and WT MCs are comparable at P0, while CAMSAP2 deficient MCs have more intersections at P7 (Fig. 4C–F). Besides, *Camsap2* deletion also decreases the proportion of MCs with single crossings at P7 (Fig. 4G), and CAMSAP2 deficient MCs tend to project into glomeruli that are further away in distance (Figs. 3I and 4H). Even though MCs at P0 do not show any differences, we still cannot completely exclude the possibility that CAMSAP2 regulates the growth of dendrites before remodeling (Yau et al, 2014), as rAAV virus usually needs weeks to start its expression, which may miss the key stage of dendritic extension. However, *Camsap2*(−/−) mice show growth delay at postnatal day (Appendix Fig. S1E), suggesting that CAMSAP2 may have stronger function during the remodeling stage. Thus, our results indicate that CAMSAP2 is required for the remodeling process of MCs.

In this study, we reveal that CAMSAP2 is required for the male mice mating and olfaction. Furthermore, we found that CAMSAP2 participates in the morphogenesis of MC dendrites. Surprisingly, our group's newly published work demonstrates that CAMSAP1 but not CAMSAP2 plays an essential role in male fertility via regulating the process of spermiogenesis (Hu et al, 2023). Thus, our studies in the nervous and reproductive systems suggest that mammalian development requires the sophisticated cooperation of CAMSAP family members. Thus, finely tuned MT networks regulated by CAMSAPs are critical for the survival of mammals.

Each type of OSN typically projects its axons to two glomeruli in the OB, and MCs in the OB extend a single apical dendrite to a glomerulus to establish the connection between OR and olfactory cortex in a spatiotemporal manner (Imamura et al, 2011; Mombaerts et al, 1996). In this study, we observed that knocking out *Camsap2* resulted in abnormal morphology of MC dendrites, including multiple apical dendrites or no apical dendrites extended into EPL. All of these defects would be predicted to disrupt the wiring of the olfactory circuit and impair the detection or discrimination of chemical cues. In the OB, each glomerulus is innervated by apical dendrites from ~20 MCs (Imai, 2014; Ravi et al, 2017). These MCs constitute a group of sister MCs and code information about odor in a non-redundant manner (Dhawale et al, 2010). Though half of MCs had single apical dendrites when *Camsap2* was knocked out in those neurons, it was apparent that the projections of their apical dendrites were severely disrupted (Fig. 3F,I,J). This aberration leads to some adjacent MCs receiving different input while some non-adjacent MCs receive the same input, and these organizational mistakes may be harmful to odor information processing and eliciting behavior. Thus, the defects in the morphogenesis of MCs may contribute to the olfactory deficits observed in *Camsap2* KO mice, but that other (as yet unassessed) differences in the KO olfactory system may also play an important role. In our future work, we will try to design more experiments to explain it.

How do the MC dendrites find the right glomerulus to construct the precise circuit? Recent works show that Rac1 and RhoA regulate the MC dendrite remodeling in an activity-dependent manner (Aihara et al, 2021; Fujimoto et al, 2023). Interestingly, our previous study showed that CAMSAP3, the homolog of CAMSAP2, inhibits the activity of RhoA via GEF-H1 and this function is related to the detyrosination of MTs (Nagae et al, 2013). Since CAMSAP2 and CAMSAP3 are both responsible for the detyrosination of MTs (Tanaka et al, 2012), we propose that CAMSAP2 may responds to some signal to remodel dendrites through regulating the activity of RhoA. However, this hypothesis needs further investigation. *Camsap2* KO mice have thinner EPL and some of MCs lack connection with glomerulus, these hint that CAMSAP2 participates in multiple processes of dendrite remodeling, such as growth, pruning and stabilizing. Furthermore, our results suggest that *Camsap2* KO mice can be used as a model to study the molecular mechanism of dendrite remodeling. Although technical limitations constrained our exploration of the molecular mechanism, we are optimistic about uncovering the intricate role of CAMSAP2 as a mediator between remodeling signaling and the MT network. The potential breakthroughs awaiting us in future research hold the promise of shedding light on this captivating relationship.

For sexually reproducing species, male and female behavioral patterns are different in copulatory stages, with mounting, intromission and ejaculation performed by the male. For this reason, *Camsap2*(-/-) females could successfully birth pups in the mating assay. *Camsap2* KO mice did not display any normal innate responses to pheromonal and non-pheromonal odorants, suggesting that the function of the main olfactory bulb and the accessory olfactory bulb are both impaired. Furthermore, we found that deletion of *Camsap2* resulted in significantly thinner EPL in all six regions of the OB (Fig. EV2A–C). This suggests that CAMSAP2 may function in the remodeling of MCs in all areas other than some specific regions of OB. If this is the case, one would expect that the deletion of *Camsap2* would cause total loss of general olfaction ability. Indeed, our observations support this conclusion (Fig. 2). Thus, our results demonstrate a vital role of CAMSAP2 in the development of the olfactory system.

## Methods

### Animals

All animal experiments were approved by the Animal Center of the Institute of Genetics and Developmental Biology (IGDB), Chinese

Academy of Sciences, and were conducted in accordance with the IACUC guidelines at IGDB. Mice were maintained in specific pathogen-free conditions in the animal facility and housed on a 12-h light/12-h dark cycle with ad libitum access to food and water. The *Camsap2*(+/−) and *Camsap2*fl/+ mice models were created by Beijing Biocytogen. Nestin-Cre mouse strain was gifted by Dr. Shilai Bao (IGDB). Unless stated otherwise, mice of both genders were used and their ages were indicated in the Figures or the legends. For timed pregnancy, mid-day when the vaginal plug was identified was calculated as embryonic day 0.5 (E0.5).

## Immunofluorescence

Immunostaining tissue was performed as described before(Zhou et al, 2020), with following antibodies: anti-CAMSAP2 (1:500, Proteintech, 17880-1-AP), anti-Tbr2 (1:1000, Abcam, ab183991), anti-NeuN (1:1000, Abcam, ab104224), anti-NF165 (1:300, DSHB, 2H3), anti-GFP (1:500, Invitrogen, A10262; 1:500, MBL, 598), anti-RFP (1:500, Chromotek, 5F8), anti-Pax6 (1:1000, BioLegend, 901301), anti-BrdU (1:1000, Abcam, ab6326), anti-Cleaved-Caspase 3 (1:800, CST, 9661), anti-PGP9.5 (UCH-L1, 1:200, Santa Cruz, sc-271639). Alexa fluor-conjugated secondary antibodies (Invitrogen) were used at 1:1000 dilution. For some antibodies, eg. Tbr2, the antigen retrieval was performed by steaming samples in a 98 °C antigen retrieval solution for 5 min and leaving samples in antigen retrieval solution to cool down to room temperature, the subsequent procedure was performed as described before (Zhou et al, 2020).

For BrdU labeling, pregnant females received a single intraperitoneal injection of BrdU (Sigma-Aldrich, 50 µg/g) at E11.5. Mice were sacrificed after injection for 2 h and fixed with PFA. The cryosections of the brain were incubated in HCl for appropriate times to denature DNA and then neutralized in 0.1 M borate buffer (pH 8.5). Subsequently, the section was immunostained with anti-Pax6 and BrdU antibodies following the standard protocol.

## Clearing with BABB

PFA-fixed brains were rinsed in PBS and embedded in 3% agarose gel. Samples were then sliced by a microslicer (Compresstome VF-300-0Z, Precisionary) at 300–400 µm thickness for P7 (Fig. 4D) and P21-P23 (Fig. 3F) mice. The clearing procedure was described previously (Erturk et al, 2011). Briefly, tissues were successively incubated in 50% and 80% tetrahydrofuran (THF) for 30 min. After three times in 100% THF for 30 min each, tissues were incubated in 100% dichloromethane for 20 min. Finally, the tissues were submerged in a mixture of BABB (benzyl alcohol and benzyl benzoate at a ratio of 1:2), as the tissues are thin, they were transparent in minutes and the tissue could be mounted on slides with BABB for confocal imaging.

To get higher signal of mScarlet positive (mScarlet+) neurons, we used the following protocol to stain the tissue with an anti-RFP antibody. The OB sections were permeabilized with wash buffer (1% Triton X-100 in PBS) three times for 30 min each, and after blocking with blocking solution (10% FBS and 5% BSA in wash buffer) O/N at 4 °C, the sections were incubated with primary antibodies at 4 °C for 2–4 days on a shaker. Then, the sections were washed with wash buffer and incubated with secondary antibody for 1–2 days on a shaker at 4 °C. Finally, the sections were washed

with wash buffer, and washed with PBS three times after fixing with 4% PFA at RT for 20 min.

## H&E and Golgi-cox staining analysis

Testes were dissected from adult mice and fixed in 4% PFA O/N at 4 °C with rocking. After washing three times with PBS, the testes were stored in 70% ethanol until the process for paraffin, and the paraffin sections were placed on slides for H&E staining analysis.

For Golgi-cox staining, 2 months brains were stained using FD Rapid Golgi Stain Kit (FD Neurotechnologies PK401A) according to the manufacturer's suggestions. OBs were sectioned at 100 µm and placed onto gelatin-coated microscope slides.

## Immunoblotting

Immunoblotting was performed as described before (Zhou et al, 2020), with the following antibodies: anti-CAMSAP1 (1:1000, Sigma, HPA024161), anti-CAMSAP2 (1:1000, Sigma, HPA027302), anti-CAMSAP3 (1:1000, homemade), anti-tubulin (1: 20,000, Sigma, T6074).

## Hormones

To generate estrus females, bilateral ovariectomy was carried out in 6–8 weeks-old female C57BL/6N mice. Ovariectomized females were allowed to recover for 3 weeks, the estradiol benzoate was subcutaneously injected at 48 h (10 µg in 100 µL sesame oil) and 24 h (5 µg in 50 µL sesame oil) before experiments, and progesterone (50 µg) was subcutaneously injected 4–6 h before experiments.

For analyzing the titers of serum sexual hormones, orbital blood was collected from adult males using a capillary tube, and the serum testosterone and estradiol levels were measured by Beijing North Institute of Biotechnology Co., Ltd.

## Infertility characterization

We first analyzed the male fertility by counting the pups produced by males. Briefly, sexually mature male mice were caged with 2-month-old WT C57BL/6N females for 4 months, the number of litters was counted. Plugging is also defined as a phenotype of male fertility, as it indicates mice mating behavior. WT males were paired with WT or *Camsap2* KO females, *Camsap2* KO males were paired with WT females, *Camsap2*fl/fl, and *Camsap2*cKO males were paired with WT females for 5–6 days, and the days for appearance of the copulatory plug were counted. We caged the male with females at night and separated them in the morning.

Sperm were isolated from the epididymis and incubated in 37 °C pre-warmed PBS. Sperm motility was assessed using computer-assisted sperm analysis (Version.12 CEROS, Hamilton Thorne Research, Beverly, MA, USA). IVF was performed with 4-week-old C57BL/6N female, mice were superovulated by intraperitoneal (i.p.) injection with 10 IU pregnant mare serum gonadotropin (PMSG) followed 48 h later by 10 IU human chorionic gonadotropin (HCG). Cumulus-intact eggs were released from the oviducts 14–16 h after HCG injection and transferred to 200 µL drops of HTF medium. Sperm were obtained from the cauda epididymides of adult WT and *Camsap2* KO littermates and capacitated for 1 h in 200 µL drops of HTF medium. 5 µL sperm were added to the

fertilization drop, and fertilization was allowed to proceed for 6 h. After washing out unbound sperm, the eggs were incubated O/N at 37 °C in a humidified atmosphere of 5% $CO_2$. The inseminated eggs were examined for progression to the 2-cell stage, and the 2-cell embryos were transferred into the oviducts of pseudopregnant recipients.

## Behavioral assays

In all, 2–4-month-old male mice were used in behavioral assays.

### Mating behavior

A mating behavior assay was initiated at least 1 h after the onset of the dark cycle, and the behaviors were recorded using an infrared-sensitive video recorder. The estrus female was introduced to the home cage of WT or *Camsap2* KO male, and videotaping for 30 min. The mating behavioral assay was repeated three times with different females at least three days apart. Mating behavior was scored according to the following criteria: nose-to-face, and nose-to-body are collectively as "non-anogenital". Nose-to-anogenital contact is specifically defined as "anogenital". "Mount" is scored when male places its forelimbs on the back of the estrus female.

### Rotarod test

Mice were first acclimated to the rotarod for 2 min, followed by accelerating from 4 to 40 rpm, with the maximal test duration set at 5 min. The test was performed for consecutive 2 days and four trials per day with at least 60 min of rest between trials. The latency to fall off the rod was recorded.

### Recording of USV

USV was detected using a condenser microphone (UltraSoundGate CM16, Avisoft Bioacoustics) that was connected to a converter (Avisoft Bioacoustics). The acoustic signals were recorded and analyzed by the software (Avisoft SASLab Pro). The vocalization of the single male was recorded for 2 min as background, and the vocalization was recorded for another 5 min after introducing the female. For young pups, they were separated from their parents and recorded for 5 min.

### Urine preference test

Urine was collected from group-housed C57BL/6 N males or females. Urine samples from ten animals were pooled and stored at −80 °C. Before testing each day, aliquots of urine were thawed and diluted threefold for use. The subject mice were habituated to the clean cage for 30 min with two empty boxes on both sides, then 60 μL urine was dropped into the boxes and recorded for 5 min for analysis.

### Territorial marking assay

Male mice were placed individually in cages for 3 days to habituate. After this habituation period, a piece of filter paper was placed on the bottom of the cage for 10 min to establish baseline urine marking. Removing the filter paper and placing a new paper under the cage. 100 μL of female urine was spotted into the center of the paper and left for 10 min. The paper was treated with ninhydrin spray and air dried until urine spots developed a purple color. Papers were scanned and the percentage of the paper marked by urine was determined using Image J software.

### Buried food test

Male mice were deprived of food for 24 h and then placed in a new cage with sterile bedding. In total, 2 g of standard mouse chow was hidden from view under ~5 cm of bedding. We measured the time required for mice to find and begin to eat the food pellet. This experiment was performed in the daytime (7:00–19:00).

### Defensive behavior

Defensive behavior was performed in a rectangle $60 \times 40$ cm open-field box. Two baffles with a $10 \times 6$ cm arch in the bottom to allow the mice entrance were inserted and divided the box into three chambers. Three sides of the hiding chamber were covered by black film and the remaining side was transparent to enable filming of mouse activity. Mice were habituated to the arena for 2 days. On day 3, a rat was deeply anesthetized by i.p. injection of sodium pentobarbital and placed in the rat chamber with its paws and snout facing the wall (see in Fig. 2D). Mice spent in each chamber were counted, and the total time of this experiment is 30 min. Avoidance behavior was defined as the amount of time mice spent inside the hiding chamber. Risk assessment used in this study is the flat-back/stretch-attend response. Initially, the mouse slowly approaches the rat with a lower head, then flattens its back while nearing the rat, resulting in an extended posture with a thrusted head; mostly, this behavior accompanies retreating/fleeing behavior.

## AAV injection

AAV injection was performed essentially as in utero electroporation described previously (Zhou et al, 2020). Briefly, mice were deeply anesthetized with sodium pentobarbital, and virus mixture (AAV2/9-TRE-Cre-WPRE, $2 \times 10^9$ v.g./mL; AAV2/9-hSyn-DIO (tTA-P2A-mScarlet)-WPRE, $2 \times 10^{13}$ v.g./mL) was injected into the embryonic OB at E14.5 with a glass microcapillary. Capillary was empirically inserted into the center of the OB, and each time, we injected 0.2 μL into a single OB base on the minimum scale of the syringe.

## Image processing

Images of sectioned brain slices and neurons were obtained through Zeiss confocal microscope Observer Z1 equipped with a spinning disk (Yokogawa CSU-1), confocal microscope LSM800, Olympus FV3000 confocal microscope, and Nikon confocal microscope (Eclipse Ti-C2). For H&E staining, images were obtained through Nikon Eclipse 80i equipped with DS-Ri1 microscope camera. For Golgi-cox staining, images were obtained via Olympus BX51 microscope.

## Statistical analyses

All the statistical details of experiments can be found in the figure legends. Graphpad Prism and Microsoft Excel were used for statistical analysis. All data collection and analysis for behavioral assays were blinded. The analysis for tissue sections was not blinded as the structural differences between WT and KO or cKO mice OB is so significant and it is easy to distinguish (narrowed EPL and blurred boundary of MCL). All the data are represented as mean ± SEM, and all conditions statistically different from the control are indicated

(*$P < 0.05$, **$P < 0.01$, ***$P < 0.001$; n.s. not significant, $P > 0.05$). All experiments were performed with at least three biological replicates.

For normal immunostaining, at least two serial sections in the medial of OB were collected. For Golgi-cox staining (Fig. 3C) and BABB clearing tissue (Figs. 3F and 4D), all sections from the OB were collected. All images of immunostained tissue were analyzed using image J to quantify the cell number, morphology, and length of EPL manually.

## Data availability

This study includes no data deposited in external repositories. All data generated during this study are included in the main figures, expanded view, and appendix.

The source data of this paper are collected in the following database record: biostudies:S-SCDT-10_1038-S44319-024-00166-x.

## Peer review information

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

## Acknowledgements

The authors thank Weixiang Guo and Shilai Bao for advice. The authors thank members of the Meng group for helpful discussions. This work was supported by the National Natural Science Foundation of China (31930025 to WM, 32100760 to ZZ, 32270736 to WM), National Key Research and Development Program of China (2018YFA0801104 and 2021YFA0804802 to WM), SUMC Scientific Research Initiation Grant (510858072 to ZZ).

## Author contributions

**Zhengrong Zhou**: Conceptualization; Data curation; Formal analysis; Funding acquisition; Investigation; Visualization; Methodology; Writing—original draft. **Xiaojuan Yang**: Conceptualization; Data curation; Formal analysis; Investigation; Visualization; Methodology. **Aihua Mao**: Investigation; Visualization; Methodology. **Honglin Xu**: Data curation; Visualization. **Chunnuan Lin**: Data curation; Investigation. **Mengge Yang**: Validation; Investigation. **Weichang Hu**: Methodology. **Jinhui Shao**: Investigation. **Peipei Xu**: Investigation. **Yuejia Li**: Validation. **Wenguang Li**: Methodology. **Ruifan Lin**: Methodology. **Rui Zhang**: Investigation. **Qi Xie**: Methodology. **Zhiheng Xu**: Conceptualization; Formal analysis. **Wenxiang Meng**: Conceptualization; Data curation; Formal analysis; Supervision; Funding acquisition; Investigation; Visualization; Methodology; Writing—review and editing.

Source data underlying figure panels in this paper may have individual authorship assigned. Where available, figure panel/source data authorship is listed in the following database record: biostudies:S-SCDT-10_1038-S44319-024-00166-x.

## Disclosure and competing interests statement

The authors declare no competing interests.

# Expanded View Figures

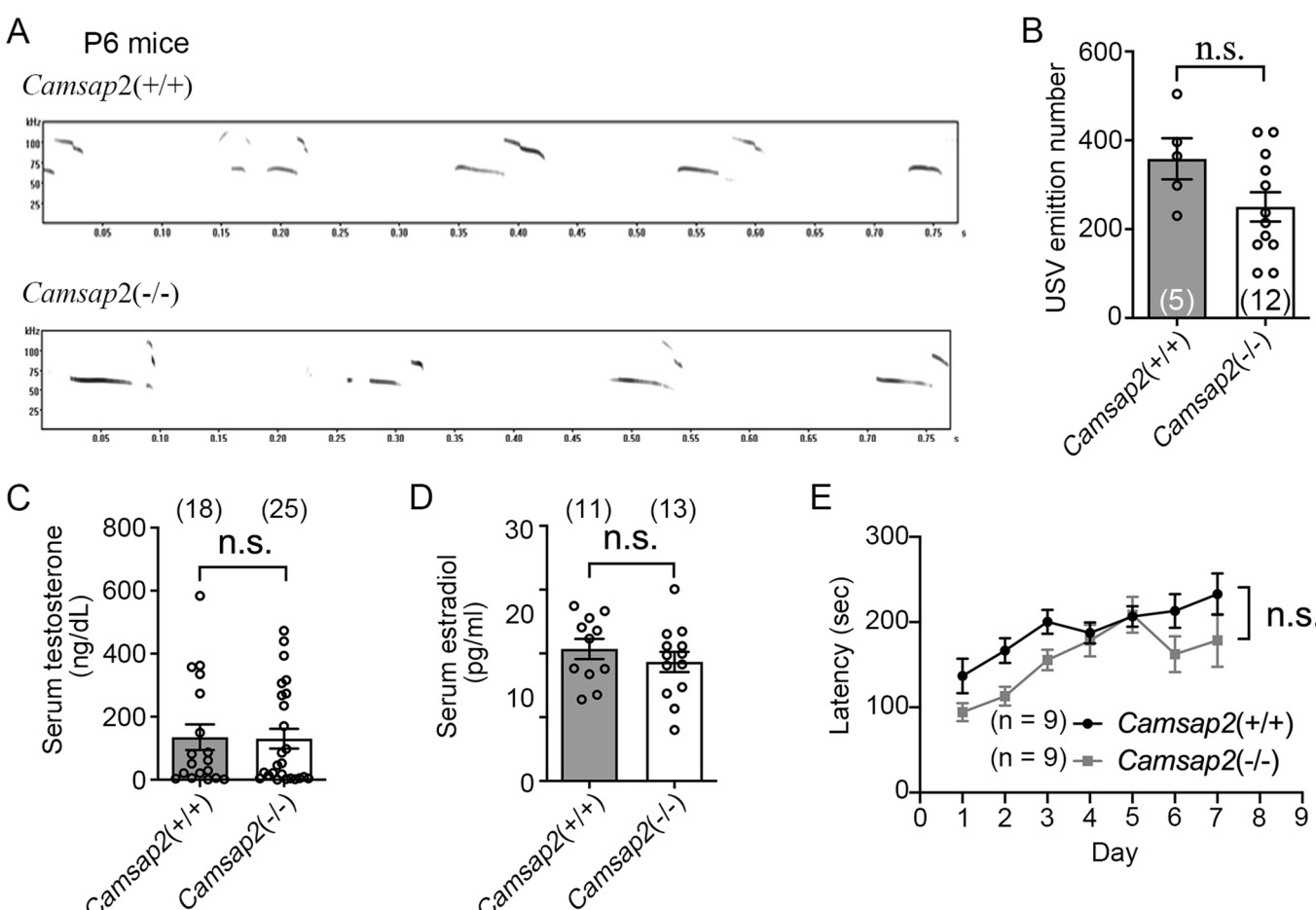

**Figure EV1. Hormone levels and motor function are not responsible for the infertility of *Camsap2*(−/−) male mice.**

(A) Representative spectrograms of USVs emitted by P6 mice. (B) Quantification of the number of USVs emitted by P6 mice (unpaired Student's *t* test, *n* = 5 or 12 biological replicates). (C) The concentration of serum testosterone (unpaired Student's *t* test, *n* = 18 or 25 biological replicates). (D) The concentration of serum estradiol (unpaired Student's *t* test, *n* = 11 or 13 biological replicates). (E) Motor performance on an accelerating rotarod, no significant difference exists at any time point (two-way R-M ANOVA, *n* = 9 biological replicates). Data information: Data are represented as mean ± SEM. n.s. not significant, *P* > 0.05.

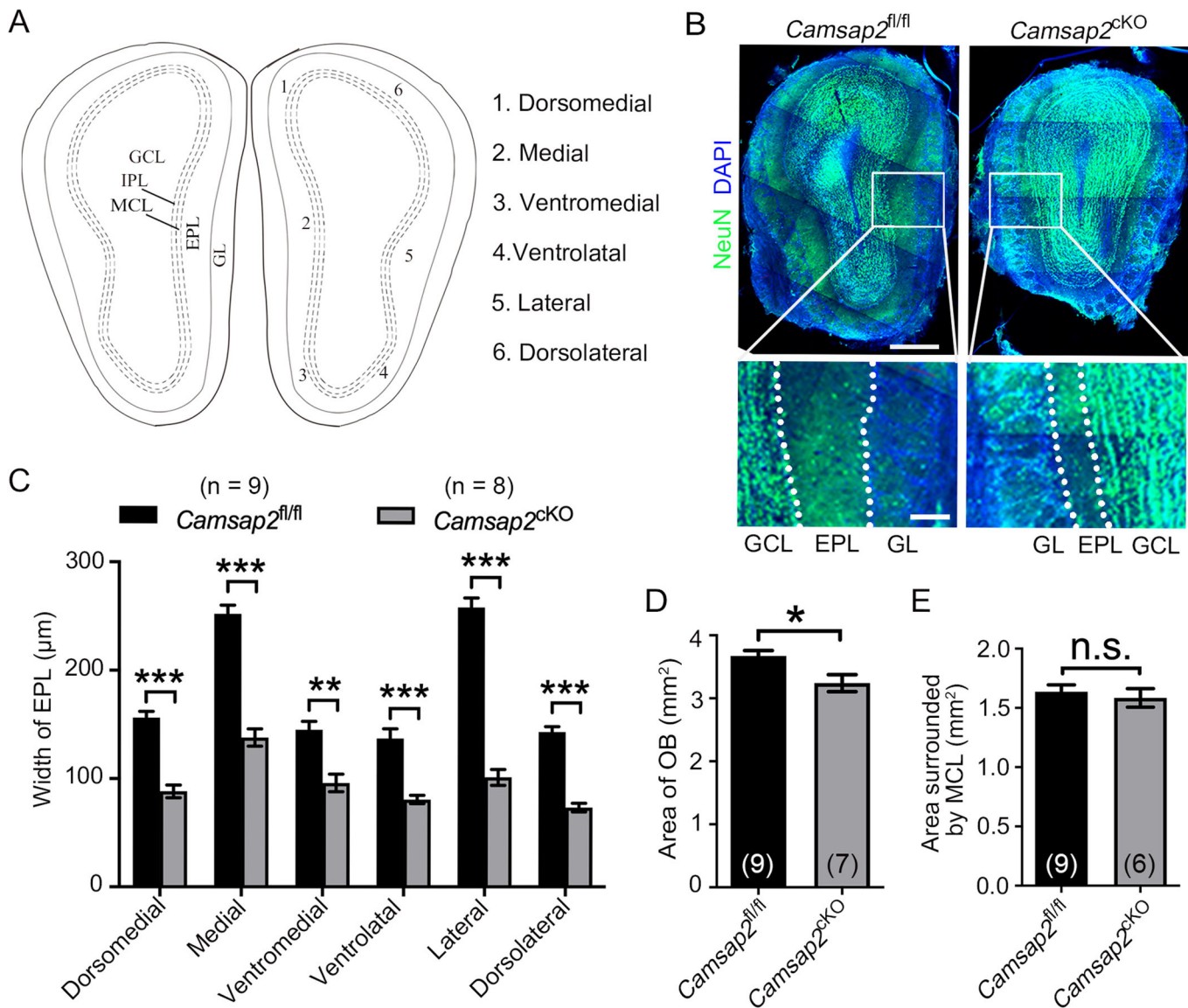

**Figure EV2. Knockout of *Camsap2* diminishes the thickness of EPL in the OB.**

(A) Schematic diagram of the OB. (B) Representative images of the OB from *Camsap2*[fl/fl] and *Camsap2*[cKO] mice, mature neurons were immunostained with antibody anti-NeuN. (C) Quantification of the thickness of EPL from *Camsap2*[fl/fl] and *Camsap2*[cKO] mice (two-way R-M ANOVA with Šídák's multiple comparisons test, $n = 3$ biological replicates, ***$P = 0.0009$ or $< 0.0001$, **$P = 0.0033$). (D) Quantification of the area of the OB from *Camsap2*[fl/fl] and *Camsap2*[cKO] mice (unpaired Student's *t* test, $n = 3$ biological replicates, *$P = 0.0221$). (E) Quantification of the area surrounded by MCL from *Camsap2*[fl/fl] and *Camsap2*[cKO] mice (unpaired. Student's *t* test, $n = 3$ biological replicates). Data information: Data are represented as mean ± SEM. n.s. not significant, $P > 0.05$. Scale bars: 400 μm and 100 μm in full-size images and zoomed areas.

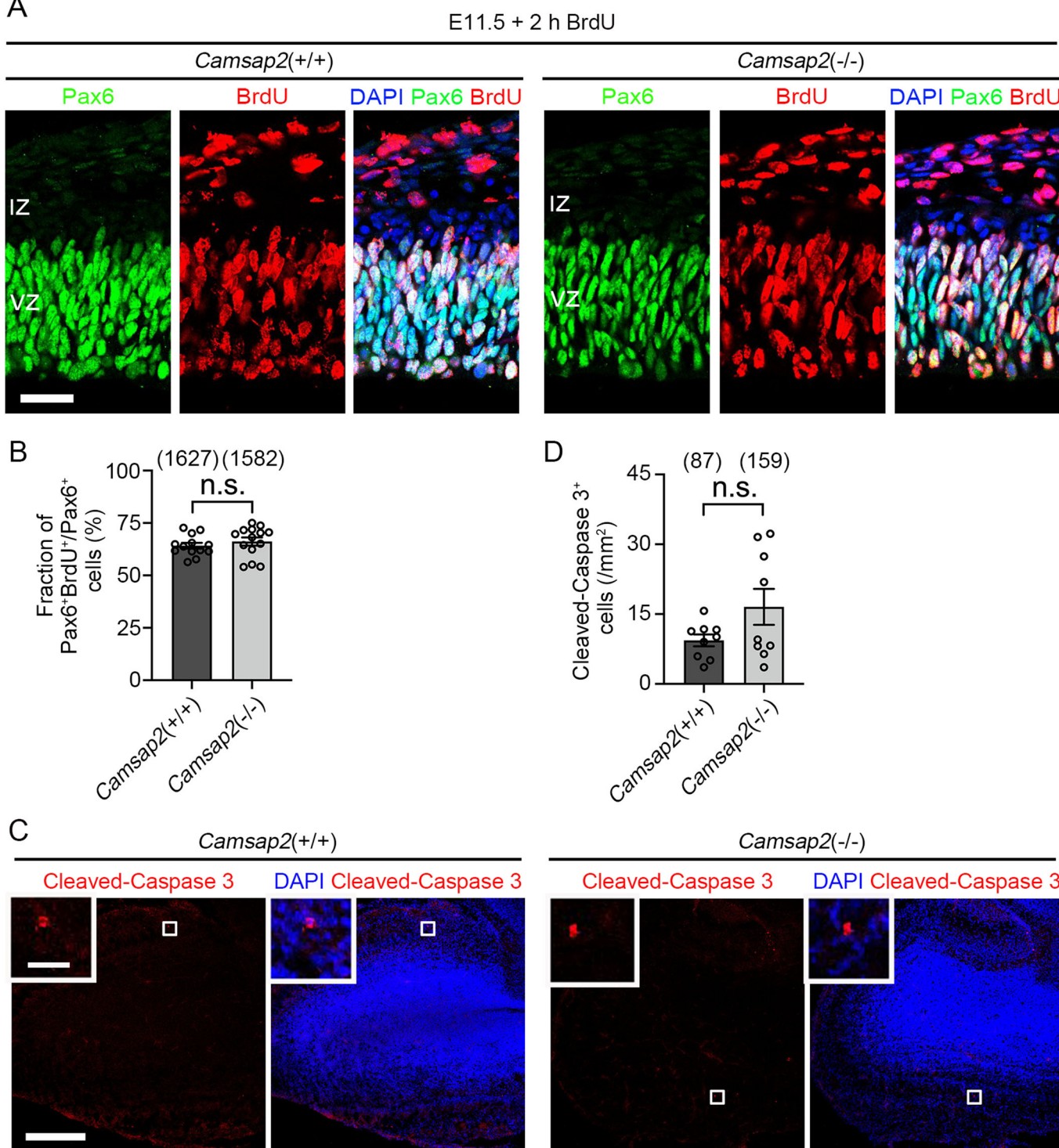

**Figure EV3. CAMSAP2 is dispensable for the neurogenesis and apoptosis of MCs.**

(A) Representative images of coronal OB sections from E11.5 mice. The progenitor cells and proliferating. cells are immunostained with antibodies against Pax6 and BrdU after BrdU labeling for 2 h. (B) The percentage of proliferating progenitor cells (unpaired Student's *t* test, *n* = 4 biological replicates). (C) Representative images of sagittal OB sections from P0 mice. The cells undergoing apoptosis are immunostained with antibody against Cleaved-Caspase 3 (Activated-Caspase 3). (D) The density of Cleaved-Caspase 3+ cells in OB (unpaired Student's *t* test, *n* = 3 biological replicates). Data information: Data are represented as mean ± SEM. n.s. not significant, *P* > 0.05. Scale bars: (A), 30 μm; (C) 300 μm and 30 μm in full-size images and zoomed areas.

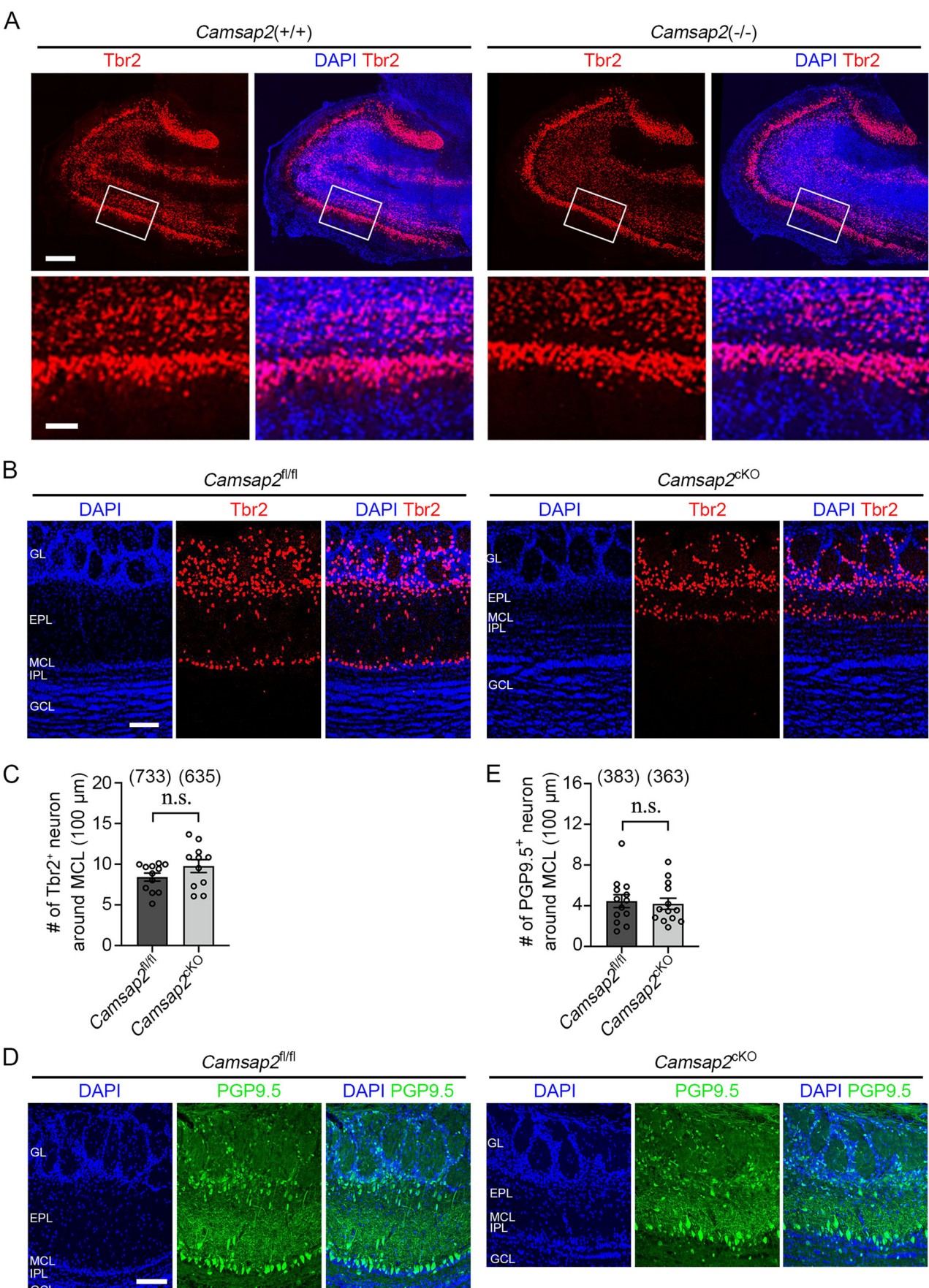

◀ **Figure EV4. CAMSAP2 is dispensable for the migration of MCs.**

(**A**) Representative images of the sagittal sections of the OB from P0 mice. The MCs are immunostained with antibody against Tbr2. (**B**) Representative images of coronal sections of the OB of adult mice. The MCs are immunostained with. antibody against Tbr2. (**C**) Quantification of the number of Tbr2$^+$ neurons around the MCL (unpaired Student's *t* test, $n = 3$ biological replicates). (**D**) Representative images of coronal sections of the OB of adult mice. The MCs are immunostained with. antibody against PGP9.5. (**E**) Quantification of the number of PGP9.5 positive neurons around the MCL (unpaired Student's *t* test, $n = 3$ biological replicates). Data information: Data are represented as mean ± SEM. n.s. not significant, $P > 0.05$. Scale bars: (**A**) 200 µm and 50 µm in full-size images and zoomed areas; (**B, D**) 400 µm.

