## [Peer Review File · EMBO Reports]

Deficiency of CAMSAP2 impairs olfaction and the morphogenesis of mitral cells

Wenxiang Meng, Zhengrong Zhou, Xiaojuan Yang, Aihua Mao, Honglin Xu, Chunnuan Lin, Mengge Yang, Weichang Hu, Jinhui Shao, Peipei Xu, Yuejia Li, Wenguang Li, Ruifan Lin, Rui Zhang, Qi Xie, and Zhiheng Xu

Corresponding author(s): Wenxiang Meng (wxmeng@genetics.ac.cn), Zhengrong Zhou (zrzhou@genetics.ac.cn)

Review Timeline:

Submission Date:	6th Oct 23
Editorial Decision:	9th Nov 23
Revision Received:	18th Feb 24
Editorial Decision:	3rd Apr 24
Revision Received:	7th Apr 24
Accepted:	22nd Apr 24

Editor: Deniz Senyilmaz Tiebe

Transaction Report:

Dear Prof. Meng,

Thank you for the transferring your research manuscript to our journal, which was now seen by three referees, whose reports are copied below.

The referees express interest in the proposed role of CAMPSAP2 in regulation of dendritic patterning of mitral cells and mating behavior. However, they also raise significant concerns that need to be addressed to consider publication here.

In particular,

- All referees find that significant textual editing is warranted as currently not all claims are supported by the data - i.e the data currently do not support a role for CAMSAP2 specifically in the pruning of mitral cell dendrites (referee #1). Please use 'dendritic morphogenesis (or remodeling)' instead of 'pruning of mitral cell dendrites', as suggested by referee #1. Moreover, the proposed causal links between mitral cell dendritic abnormalities, olfactory deficit and infertility are also not sufficiently supported (referee #3). Therefore, the corresponding statements need to be altered in the text.
- Some analyses need to be performed in female mice as well (referee #2, point 3).
- Significantly more details need to be provided regarding the figure legends, statistics and materials and methods (all referees).
- Additional controls are required for some experiments (all referees).

We note that referee #3 recommends removal of some of the panels due to caveats with the experiments. If you choose not to do so, please consider moving them to the Expanded View Figures and please include discussion points along the lines of the comments of referee #3.

Given these positive recommendations, we would like to invite you to submit a revised manuscript. Please revise your manuscript with the understanding that the referee concerns (as in their reports) must be fully addressed and their suggestions taken on board. Please address all referee concerns in a complete point-by-point response. Acceptance of the manuscript will depend on a positive outcome of a second round of review. It is EMBO reports policy to allow a single round of major experimental revision only and acceptance or rejection of the manuscript will therefore depend on the completeness of your responses included in the next, final version of the manuscript.

We realize that it is difficult to revise to a specific deadline. In the interest of protecting the conceptual advance provided by the work, we recommend a revision within 3 months. Please discuss the revision progress ahead of this time with me if you require more time to complete the revisions, or if you have questions or comments regarding the revision (also by video chat).

1. A data availability section providing access to data deposited in public databases is missing (where applicable).
2. Your manuscript contains statistics and error bars based on $n=2$. Please use scatter plots in these cases.

You can submit the revision either as a Scientific Report or as a Research Article. For Scientific Reports, the revised manuscript can contain up to 5 main figures and 5 Expanded View figures, and it should not exceed 27000 characters. If the revision leads to a manuscript with more than 5 main figures it will be published as a Research Article. In this case the Results and Discussion section should be separate. If a Scientific Report is submitted, these sections have to be combined. This will help to shorten the manuscript text by eliminating some redundancy that is inevitable when discussing the same experiments twice. In either case, all materials and methods should be included in the main manuscript file.

3) We replaced Supplementary Information with Expanded View (EV) Figures and Tables that are collapsible/expandable online. A maximum of 5 EV Figures can be typeset. EV Figures should be cited as 'Figure EV1, Figure EV2' etc... in the text and their respective legends should be included in the main text after the legends of regular figures.

- For the figures that you do NOT wish to display as Expanded View figures, they should be bundled together with their legends in a single PDF file called *Appendix*, which should start with a short Table of Content. Appendix figures should be referred to in

the main text as: "Appendix Figure S1, Appendix Figure S2" etc. See detailed instructions regarding expanded view here: <https://www.embopress.org/page/journal/14693178/authorguide#expandedview>

4) a .docx formatted letter INCLUDING the reviewers' reports and your detailed point-by-point responses to their comments. As part of the EMBO publication's Transparent Editorial Process, EMBO reports publishes online a Review Process File (RPF) to accompany accepted manuscripts. This File will be published in conjunction with your paper and will include the referee reports, your point-by-point response and all pertinent correspondence relating to the manuscript.

<https://www.embopress.org/page/journal/14693178/authorguide#transparentprocess>

5) a complete author checklist, which you can download from our author guidelines

<https://www.embopress.org/page/journal/14693178/authorguide>. Please insert information in the checklist that is also reflected in the manuscript. The completed author checklist will also be part of the RPF.

6) Please note that all corresponding authors are required to supply an ORCID ID for their name upon submission of a revised manuscript (<https://orcid.org/>). Please find instructions on how to link your ORCID ID to your account in our manuscript tracking system in our Author guidelines

<https://www.embopress.org/page/journal/14693178/authorguide#authorshipguidelines>

7) Before submitting your revision, primary datasets produced in this study need to be deposited in an appropriate public database (see <https://www.embopress.org/page/journal/14693178/authorguide#datadeposition>). Please remember to provide a reviewer password if the datasets are not yet public. The accession numbers and database should be listed in a formal "Data Availability" section placed after Materials & Method (see also

<https://www.embopress.org/page/journal/14693178/authorguide#datadeposition>). Please note that the Data Availability Section is restricted to new primary data that are part of this study. * Note - All links should resolve to a page where the data can be accessed. *

Additional information on source data and instruction on how to label the files are available:

<https://www.embopress.org/page/journal/14693178/authorguide#sourcedata>

9) Our journal encourages inclusion of *data citations in the reference list* to directly cite datasets that were re-used and obtained from public databases. Data citations in the article text are distinct from normal bibliographical citations and should directly link to the database records from which the data can be accessed. In the main text, data citations are formatted as follows: "Data ref: Smith et al, 2001" or "Data ref: NCBI Sequence Read Archive PRJNA342805, 2017". In the Reference list, data citations must be labeled with "[DATASET]". A data reference must provide the database name, accession number/identifiers and a resolvable link to the landing page from which the data can be accessed at the end of the reference. Further instructions are available at <http://www.embopress.org/page/journal/14693178/authorguide#referencesformat>

10) Regarding data quantification (see Figure Legends:

<https://www.embopress.org/page/journal/14693178/authorguide#figureformat>)

- the name of the statistical test used to generate error bars and P values,
- the number (n) of independent experiments (please specify technical or biological replicates) underlying each data point,
- the nature of the bars and error bars (s.d., s.e.m.),
- If the data are obtained from n Program fragment delivered error ``Can't locate object method "less" via package "than"

(perhaps you forgot to load "than"?) at //ejpvfs23/sites23b/embor_www/letters/embor_decision_revise_and_review.txt line 56.' 2, use scatter blots showing the individual data points.

12) Please also note our reference format:

I look forward to seeing a revised version of your manuscript when it is ready. Please let me know if you have questions or comments regarding the revision.

Kind regards,

Deniz Senyilmaz Tiebe

Deniz Senyilmaz Tiebe, PhD
Editor
EMBO Reports

Referee #1:

In this study, the authors comprehensively characterized Camsap2 KO mice. First, they found that male CAMSAP2 KO mice are infertile. By carefully analyzing the KO mice, they found that the male KO mice are defective in olfactory-guided mating behavior. While odor responses in the OSNs were preserved, responses in the OB and olfactory cortex were reduced. The most prominent defects were found in the dendritic morphology of mitral cells in the OB, suggesting that CAMSAP2 is critical for dendritic patterning in mitral cells. Recent studies have shown that spontaneous neuronal activity and molecular factors (e.g., Notch and BMP signaling) control the dendritic morphogenesis in mitral cells. Therefore, CAMSAP2 may be involved in these mechanisms. On the other hand, the current study reveals only a little about how microtubule dynamics contribute to dendritic morphogenesis. Furthermore, although the authors describe pruning defects, this alone cannot explain the defective mating behavior. It seems that dendritic growth is also affected in the KO mice.

Overall, this is a massive piece of work. I believe that the authors have spent a considerable amount of time and effort on this study. I suggest that this study be published without additional sets of experiments. However, the authors need to be more careful in the presentation of the data and the interpretation of the results.

Major comments:

From the data presented, it is unlikely that CAMSAP2 is specifically involved in the pruning of mitral cell dendrites. Pruning defects alone cannot explain the defective odor responses in OB neurons shown in Fig. 3. In Fig. 4 and 5, there is a significant fraction of mitral cells that lack any connection to the glomerulus, suggesting that dendritic growth or stabilization is also affected. The EPL is much thinner in the KO, suggesting that lateral dendrite formation is also affected. I, therefore, speculate that CAMSAP2 may be involved in more general aspects of dendritic remodeling and/or stability. I suggest the authors use "dendritic morphogenesis (or remodeling)" instead of "pruning" in the title. The main text also needs to be edited accordingly.

Recent studies (Aihara et al., Cell Rep 2021; Fujimoto et al., Dev Cell 58:1221-1236, 2023) indicate that Rac1/RhoA-mediated regulation of the actin cytoskeleton is primarily important for activity-dependent dendrite remodeling/pruning in mitral cells. Possible roles of microtubule dynamics should be discussed in this context.

Minor comments:

Fig. S1. In general, anosmic mice show suckling defects and severe growth delays during the neonatal stages. In contrast,

Camsap2 KO shows olfactory defects at later stages, suggesting that Camsap2 may be necessary for maintaining mitral cell connectivity.

Fig. 2. Figure legends should clearly indicate that panels G-J are cKO data.

Line 184. pS6 is not an IEG. See Knight et al. for details.

Fig. 3. In the pS6 and c-fos immunostaining experiments, negative controls without odor stimulation are required because there should be background staining signals without odor stimulation. The authors also need to mention the caveat that pS6 and IEG are indirect readouts of neuronal activity (unlike Ca²⁺ imaging or electrophysiology).

Fig. 4. Very similar defects in dendrite patterning have been reported for Tbr2 KO. See Figure 4 in Mizuguchi et al., J Neurosci 32(26):8831-8844 (2012).

Line 238. CAMSAP2 is required for normal dendrite development, but I don't think CAMSAP2 "instructs" MC development. Most likely, CAMSAP2 only plays a permissive role in wiring specificity.

Fig. 5I shows that CAMSAP2 interacts with Katanin P60L1. However, there are no functional data for Katanin P60L1 in mitral cells. It is not clear whether it is expressed in developing mitral cells. The role of Katanin P60L1 is too speculative here, and Fig. 5I should be removed, at least from the main figure.

Fig. 6. The authors need to perform a quantitative evaluation of the KO efficiency for AAV experiments, as was done for Tbx21-iCre.

Line 321. Without sparse labeling with AAV, it is difficult to conclude whether mitral cells have single primary dendrites.

Fig. 7D. Without quantification, it is difficult to conclude that Camsap1(-/-) has defects in migration.

Line 373. A model proposed by Nishizumi et al. is inconsistent with an earlier study based on more accurate and quantitative tracing. See Figure 5-6 in Ke et al., Nat Neurosci 16(8):1154-1161 (2013).

Referee #2:

This study evaluates the role of CASMAPs protein family in the development and function of olfactory circuits, with a particular focus on CASMAP2. The authors developed different genetic models to knock out the protein of interest. The absence of CASMAP2 led to an impairment in both mating behavior and basic olfactory functions in the male cohort. At cellular level, Casmap2 ^{-/-} animals exhibited abnormalities in the Olfactory Bulb (OB) anatomy. Specifically, MCs were not only altered in their morphology but presented impairments at the level of dendrite pruning, key process for the establishment of the olfactory circuitry. These alterations led to an overall decrease in the activity of the circuitry, explaining the anosmia and mating-related impairments. Additionally, the authors briefly expand on the other members of the CASMAP family. Casmap1 ^{-/-} animals showed defect in the migration of MCs while Casmap3 ^{-/-} animals did not show any impairment.

The study is interesting and well executed. The significance of this work resides in a complete characterization of the CASMAP2's role in supporting olfaction, including a behavioral, reproductive and cellular perspective. This study may provide new general insight on how relevant the microtubule network is to cognitive functions. Beside this, the Casmap2 ^{-/-} line may be considered and used as a model to study olfactory dysfunction and the role that MCs may play in that. I only have a few points of improvement to suggest.

Major points

1. The experimental work is strong and data appear consistent with the general message of the study. However, the quality of the text is generally rather poor and, at times, difficult to understand. Advice from professional support on English writing is highly advised.

2. In different sections, claims are rather strong (e.g. data demonstrate/prove/indicate). However, as long as data are consistent with the claim, they often are not directly proving it. This again may have to do with English. Few examples of this general tendency are below:

- Line 175-181: claim that no differences in position and number of mature OSNs and other populations in the MOE is not

supported by quantification. What is shown are qualitative images of MOE in the supplementary materials. Please tone down the claim (e.g.: "no major change was evident") or, alternatively, provide quantification.

- Line 241-250: claim that Casmap2 $-/-$ animals do not show defects in neither migration nor neurogenesis. This claim is supported by quantification of Tbr2+ and PGP9.5+ neurons at both developmental and adult stages but this alone is insufficient to directly prove "migration/neurogenesis". Combination of different BrdU paradigms for neuronal birthdating, stem cell proliferation, survival, etc., are needed to make the strong claims made by the authors.

- Other instances of strong claims are present. Authors are advised to revised the whole manuscript.

3. Across the whole study the authors focused their interest on male animals, considering their infertility and impaired mating behavior. This is absolutely fine, but it would be potentially interesting to also evaluate how and if olfactory dysfunction and cellular alterations apply to female as well. Admittedly, this is very likely, and perhaps assessment may not need to be that thorough as it was in males. For example, simple pumps recognitions by smell and/or predator avoidance would be very simple to execute.

Minor points

4. The last result section refers to KO models of CASMAP1 and CASMAP3 (line 314-332). To the reader, these results appear to be beyond the scope of this paper. The authors may consider to move them to the supplementary section, considering that the strongest point of this study remains the characterization of Casmap2 $-/-$.

5. Neither in the results section nor methods it is clearly stated how the histological quantification have been performed. Was the analysis performed stereologically collecting serial sections? Was it performed manually or automatically? Please state analysis details in the methods more clearly.

6. In different graphs (e. g. Fig. 1G, 2C, 2H) data points are poorly represented. Taking 1G as an example, the authors show that Casmap2 $-/-$ male animals do not mate over the course of 5 days. In the graph, the data points for group 2 are all indicated at Day 5 as if all mice mated on this very day. However, this is not the case. Rather, the experiment was interrupted at day 5 and these mice never mated at all (this is my understanding). Indicating points at Day 5 is misleading, rather there should be no point at all, or an interrupted bar represented and/or the y axis changes accordingly.

Referee #3:

In this extremely ambitious and wide-ranging manuscript, the authors present a series of experiments investigating the role of CAMSAP genes in the development of the mouse olfactory system. They describe a number of phenotypes resulting from knockout of CAMSAP2 or CAMSAP1, some of which are convincingly supported by the data, but many of which are not. In particular, the conclusions drawn regarding causality of effects in these mouse models are problematic. It is clear that CAMPSAP2 KO males are infertile, and that they have olfactory deficits, but it is not clear that the olfactory deficits are the cause of the infertility. More tenuous still is the proposed causal link between mitral cell dendritic abnormalities and olfactory deficits - it is absolutely not convincingly demonstrated in this paper that abnormal mitral cell dendritic development produces anosmia.

Without doing a huge amount of additional experimental work, it is not clear to me how these causal inferences could be sufficiently supported. Perhaps the most appropriate course of action would be to produce a much toned-down, descriptive version of the study which removes all of the currently uninterpretable data (current figures 3, 5I, 6 & 7), and which presents the reasonably solid findings that CAMSAP2 KO mice are infertile, that they have deficits in olfactory behaviour, and that they have abnormal mitral cell dendritic pruning, *without* attempting to draw causal links between these observations. This, in my view, could still form a useful addition to our understanding of the molecular processes underlying olfactory system development.

Major issues

- Link between infertility and olfactory deficits in the CAMSAP2 KO (Abstract line 32: 'knockout (KO) males were infertile due to impaired sense of smell'). The authors have done a thorough job of trying to identify potential sources of infertility and to narrow this down to a deficit in male copulatory behaviour. However, there is no evidence that the lack of mounting is due to an inability to detect and/or respond to olfactory cues. It is entirely possible that the CAMSAP2 KO produces this effect via deficits in other, non-olfactory circuits involved in reproductive behaviour. This is especially likely given that, despite descriptions to the contrary, there *are* motor deficits on the rotorod test, together with a strong trend (with very small sample sizes presumably responsible for non-significance) towards less ultrasonic vocalisations in the KO (FigS2). It is important here to note that the purported 'mitral-cell-specific' CAMPSAP2 KO (Fig 6), which would potentially link olfaction and sterility, has actually not been shown to be MC-specific at all. Injecting AAV anywhere into the developing embryonic brain would be expected to produce widespread

infection far from the injection site, and the authors show no evidence that M/TCs are the only cells in the nervous system affected by this manipulation. On the contrary, in fact, the labelling shown in Fig6E not only reveals off-target expression in OB granule cells (as acknowledged by the authors) but also in other upstream areas of the OB such as the glomerular layer.

- Link between mitral cell pruning and olfactory deficits in the CAMSAP2 KO. Strong claims are made about the underlying causes of olfactory deficits in this KO, e.g. Abstract line 33: 'The dysosmia was due to the pruning defect of MC and the misconnection between OSNs and MCs.' In fact, severe olfactory deficits being produced specifically by changes in mitral cell dendritic development would be an extremely surprising finding, given the wild-type accessory OB (AOB) functions extremely well with mitral cells that target multiple glomeruli. Moreover, hugely disrupted OB circuitry, including major mitral cell dendritic abnormalities, has been shown to be associated with strikingly few overt olfactory deficits (see recent publication from the Rokni lab). Here it is clear, novel, and interesting that MC dendritic pruning is affected by (cell-autonomous) CAMSAP2 KO - as stated above, I think this is useful information for the field of olfactory development. However, there is no clear evidence in this manuscript that this pruning defect underlies the severe olfactory deficits observed. See above for issues with the 'MC-specific' manipulation that is the major piece of evidence used to make this claim - the authors simply cannot claim that 'our data indicate that MC-specific deficiency of CAMSAP2 is the primary cause of dysosmia.' (line 313).
- The authors focused on the development/pruning/projection of M/Tc apical dendrites, but what about the lateral dendrites? Deficits here would be more closely related to the distribution of CAMSAP2 immunolabel shown in Fig5B, and would be far more likely than apical dendrite abnormalities to be related to the reported (convincing) reduction in EPL width.
- Issues with activity marker experiments (Fig 3, plus supplementals). There are no non-stimulation controls for baseline marker expression, no co-stains to identify the labelled cell types, no attempt to standardise labelling/imaging/analysis across conditions, and at best the most rudimentary 'quantification' involving cell counts that is not described at all in the Methods. In the *Egr1* example, for instance, it seems that the staining is weaker in the cKO, though the authors claim otherwise. I think these experiments are completely uninterpretable, and should be removed from the paper.
- Issues with the 'OSN-MC connectivity' experiment (Fig3FG). Here a single uncontrolled, highly unspecific AAV injection protocol is used to try to infer connectivity between OSNs and OB neurons. There is no demonstration that the resultant labelling in OB cells is indeed 'transsynaptic', rather than produced by a direct leak of AAV injected into (very physically close) olfactory epithelium. Even if the OB cell label here is (all) transsynaptic, there are many, many reasons why it might be different in the KO vs WT, especially when the gene involved controls microtubule function - the authors presumably assume, but cannot be sure, that AAV transport/synaptic transfer/postsynaptic uptake are all equal in the CAMSAP2 KO. This single, highly flawed approach cannot possibly be used to conclude a 'loss of synapses between OSNs and MCs'! Again, I strongly suggest that these data are removed from the paper.
- Lack of quantification throughout the paper. This includes important statements made on the basis of single qualitative examples regarding OE organisation, non-EPL OB layers/glomerular characterisation, the MC-specific KO of CAMSAP2, MCL organisation at P0 and its 'obscure' borders later, CAMSAP2 immuno, all of the CAMSAP1/3 data...
- The data in Fig 5I address a completely different question unrelated to the rest of the manuscript and do not appear to complement the wider study. I suggest these are also removed. If they are retained (with a clear rationale), there is a missing input control for CAMSAP2-BirA*-HA alone, just like conditions 1 and 2. How do we know that whatever we see in the HA blot in 3 and 4 does correspond indeed to the CAMSAP2 band? Could it be a non-specific binding of the primary? Lanes 1 and 3 in input look strangely similar in the anti GFP blot. Also, lines 271 to 273, the authors claim that 'Our results indicated that CAMSAP2 strongly interacts with Katanin P60L1, while we only occasionally found a very weak interaction between CAMSAP2 and Kif2a'. While I agree with the first statement (and even so, I would expect the CAMSAP2 band to be stronger in the pull-down conditions compared to the input, as the pull-down per se should enrich the quantity of detected protein), looking at Figure 5I, the second part is not true: if we look at the lane 4 in the pull-down panel, there is a strong band for the HA blot (indicative of the presence of CAMSAP2), but there is nothing in the GFP immunoblot. It is therefore impossible to conclude that there is an interaction from this experiment. I suggest the authors to provide an image that supports their claims, or, alternatively, remove such claim from the main text.
- Contrary to the stated conclusions (e.g. Abstract line 34), there is not a convincing MTC migratory phenotype in the CAMSAP1 KO. Instead, all that is shown is an unquantified example image of *Tbr2*⁺ cells in the GCL. How can the authors be sure that this is not just aberrant *Tbr2* expression in granule cells (or other GCL cells)? Would e.g. Golgi staining show clear MTC morphology of GCL-localised cells in the CAMSAP1 KO?

Minor issues:

- Did the controls for the 'MC-specific' KO manipulation comprise non-floxed mice that were also injected? This should have been the case. If not, all of the effects of this manipulation can probably be explained by non-specific effects of the injection procedure. But if so, why do the authors interpret their results in terms of possible dorsal OB damage (line 300)? Wouldn't similar damage have been caused by the injections in non-floxed control mice?
- Recent work from the Imai group has significantly advanced our understanding of the molecular mechanisms of mitral cell dendritic pruning. Why isn't this important and highly relevant work cited?
- All statistical tests are simple pairwise t-tests. Were assumptions for parametric tests met? Repeated tests on the same data will increase the rate of false positive 'significant' findings, so the following data (if included in any revision) would need reanalysing with appropriate tests that take multiple comparisons into consideration: Fig1G (needs 1-way ANOVA); Fig2A,B,F,J and 6D (I recommend taking ratios for each mouse, then comparing those with a t-test); Fig5F (needs 2-way ANOVA); FigS1E (needs 2-way R-M ANOVA); FigS2E (also needs 2-way R-M ANOVA); FigS6F (needs 2-way R-M ANOVA, or some kind of integrated measure per animal)
- I do not understand the reasoning on Line 628, which appears to argue that statistical tests were not performed on proportion

data because they gave non-significant results (!). Many tests are available for specifically assessing differences in proportions, and these actually tend to be rather powerful.

- The Methods section needs a lot more detail on how images were acquired, and how they were analysed. Were experimenters blind to genotype/manipulation in this study? If not, how can the authors discount implicit bias in all of their data, especially the behavioural tests?

- Scalebars are missing from many of the images.

Point-by-point response to Reviewers' comments:

We are grateful for the invaluable suggestions provided by the three reviewers for our study. Each point raised by the reviewers was given due consideration, and we have incorporated the new data based on their suggestions. As per the reviewers' recommendation, we have removed Figure 3 and Figure 6 to ensure the conclusions drawn from each experiment are convincingly supported. We also excluded Figure 7 from this version of the manuscript to keep the focus of our research on the characterization of *Camsap2*-/- in this study.

We have rewritten the manuscript to fulfill the policy of EMBO Reports, and we hope that our work will now match the standards for publication of the EMBO Reports. Below are our point-by-point responses in blue.

Referee #1:

In this study, the authors comprehensively characterized *Camsap2* KO mice. First, they found that male *Camsap2* KO mice are infertile. By carefully analyzing the KO mice, they found that the male KO mice are defective in olfactory-guided mating behavior. While odor responses in the OSNs were preserved, responses in the OB and olfactory cortex were reduced. The most prominent defects were found in the dendritic morphology of mitral cells in the OB, suggesting that CAMSAP2 is critical for dendritic patterning in mitral cells. Recent studies have shown that spontaneous neuronal activity and molecular factors (e.g., Notch and BMP signaling) control the dendritic morphogenesis in mitral cells. Therefore, CAMSAP2 may be involved in these mechanisms. On the other hand, the current study reveals only a little about how microtubule dynamics contribute to dendritic morphogenesis. Furthermore, although the authors describe pruning defects, this alone cannot explain the defective mating behavior. It seems that dendritic growth is also affected in the KO mice.

Overall, this is a massive piece of work. I believe that the authors have spent a considerable amount of time and effort on this study. I suggest that this study be published without additional sets of experiments. However, the authors need to be more careful in the presentation of the data and the interpretation of the results.

Response:

We extend our heartfelt gratitude for your insightful and constructive feedback. Your valuable input is deeply appreciated, and it greatly contributes to the enhancement of our work.

Major comments:

From the data presented, it is unlikely that CAMSAP2 is specifically involved in the pruning of mitral cell dendrites. Pruning defects alone cannot explain the defective odor responses in OB neurons shown in Fig. 3. In Fig. 4 and 5, there is a significant fraction of mitral cells that lack any connection to the glomerulus, suggesting that dendritic growth or stabilization is also affected. The EPL is much thinner in the KO, suggesting that lateral dendrite formation is also affected. I, therefore, speculate that CAMSAP2 may be involved in more general aspects of dendritic remodeling and/or stability. I suggest the authors use "dendritic morphogenesis (or remodeling)" instead of "pruning" in the title. The main text also needs to be edited accordingly.

Response:

We completely agree with the aforementioned comments. We have changed the title of our manuscript and replaced the word "pruning" in the main text with "morphogenesis", and altered the main text accordingly.

Recent studies (Aihara et al., Cell Rep 2021; Fujimoto et al., Dev Cell 58:1221-1236, 2023) indicate that Rac1/RhoA-mediated regulation of the actin cytoskeleton is primarily important for activity-dependent dendrite remodeling/pruning in mitral cells. Possible roles of microtubule dynamics should be discussed in this context.

Response:

Thank you for your advice. As the study published in Dev Cell (Fujimoto et al, 2023) was not available at the time of manuscript drafting, we were unable to cite this work

at the time.

Our previous studies suggest that CAMSAP may regulate physiological activities by regulating small G proteins such as Rac1/RhoA. In a previous study, we found that CAMSAP3 inhibits the activity of RhoA via GEF-H1 (Nagae et al, 2013). Since CAMSAP3 and CAMSAP2 are mainly localized in dendrites and axons, respectively (Zhou et al, 2020), combined with the recent studies you recommended, we have now added the possibility that CAMSAP2 controls dendrite remodeling by regulating the activity of RhoA in the Discussion section and cited these papers (lines 261-267).

Minor comments:

Fig. S1. In general, anosmic mice show suckling defects and severe growth delays during the neonatal stages. In contrast, *Camsap2* KO shows olfactory defects at later stages, suggesting that CAMSAP2 may be necessary for maintaining mitral cell connectivity.

Response:

We appreciate your suggestion. We agree with your viewpoint and have discussed the possibility in the Discussion section (lines 268-270).

Fig. 2. Figure legends should clearly indicate that panels G-J are cKO data.

Response:

Thank you for your recommendation. We have added this information in the figure legends of Figure 2 (lines 615-624).

Line 184. pS6 is not an IEG. See Knight et al. for details.

Response:

Thank you for bringing to our attention the mistake we made in the previous version of our manuscript. We apologize for the confusion caused by mixing up the classification of IEGs. We would have liked to clarify that *Egr1* and *c-fos* are IEGs, while pS6 is not, but as we mentioned below, we have decided to remove Figure 3 from the previous version of the manuscript. We plan to conduct additional experiments using a different mouse model and confirm our findings. Once we have done so, we will include these data in our next paper. Thank you again for your assistance.

Fig. 3. In the pS6 and *c-fos* immunostaining experiments, negative controls without odor stimulation are required because there should be background staining signals without odor stimulation. The authors also need to mention the caveat that pS6 and IEG are indirect readouts of neuronal activity (unlike Ca²⁺ imaging or electrophysiology).

Response:

Thank you for your professional suggestion. As mentioned previously, we have removed Figure 3 in this revised version of the manuscript. Additionally, we will include negative controls in our future works.

Fig. 4. Very similar defects in dendrite patterning have been reported for *Tbr2* KO. See Figure 4 in Mizuguchi et al., *J Neurosci* 32(26):8831-8844 (2012).

Response:

Thanks for your comments, *Camsap2* KO and *Tbr2* KO mice have similar defects, that is, narrowed EPL and enlarged angles of the apical dendrite.

Line 238. CAMSAP2 is required for normal dendrite development, but I don't think CAMSAP2 "instructs" MC development. Most likely, CAMSAP2 only plays a permissive role in wiring specificity.

Response:

Thanks for this suggestion. We realized it is incorrect to use the word "instructs" here, and we have used the word "participates" instead of the word "instructs" in the revised manuscript (Line 208 in the revised manuscript).

Fig. 5I shows that CAMSAP2 interacts with Katanin P60L1. However, there are no functional data for Katanin P60L1 in mitral cells. It is not clear whether it is expressed in developing mitral cells. The role of Katanin P60L1 is too speculative here, and Fig. 5I should be removed, at least from the main figure.

Response:

Thanks for your suggestion. We have removed Figure 5I in this revised version of the manuscript.

Fig. 6. The authors need to perform a quantitative evaluation of the KO efficiency for AAV experiments, as was done for *Tbx21-iCre*.

Response:

In the previous version of the manuscript, we evaluated the KO efficiency by analyzing the morphology of mitral cells (former Figure 6E, higher-magnification images), as deficiency of CAMSAP2 alters the morphology of mitral cells. However, we agree with this reviewer that immunostaining for CAMSAP2 with cultured neurons can provide more convincing results.

After the discussion with our co-authors, we decided to accept Reviewer 3's suggestion to remove Figure 6 from the previous manuscript. In our future study, we will specifically knock out the expression of *Camsap2* in mitral cells with another Cre tool in mice.

Line 321. Without sparse labeling with AAV, it is difficult to conclude whether mitral cells have single primary dendrites.

Response:

In the beginning, we have considered analyzing the morphology of mitral cells in *Camsap1* KO mice. However, our previous study reminded us that it is difficult to obtain this result, as we found it very difficult to get a live embryo of *Camsap1* KO after *in-utero electroporation* at E14.5. Besides, more than 50% of *Camsap1* KO mice will die in the first 2 days after birth. In this revised manuscript, we have removed the

data of CAMSAP1 to focus our research on CAMSAP2.

However, we agree with the view of this reviewer; in the future study, we will try to specifically knockout the expression of *Camsap1* in mitral cells via injection of the AAV into *Camsap1*^{flox/flox} mice other than the *Camsap1* KO mice mentioned above.

Fig. 7D. Without quantification, it is difficult to conclude that *Camsap1*(-/-) has defects in migration.

Response:

Thanks for this suggestion. As mentioned above, we have removed the data of *Camsap1*(-/-) mice.

Line 373. A model proposed by Nishizumi et al. is inconsistent with an earlier study based on more accurate and quantitative tracing. See Figure 5-6 in Ke et al., Nat Neurosci 16(8):1154-1161 (2013).

Response:

We appreciate the opportunity to revisit the models proposed by Nishizumi et al. in light of your insightful comments. After a thorough examination of the cited literature and collaborative consultations with our co-authors, we recognize that the model by Nishizumi et al. may now be viewed in the context of emerging research.

Nishizumi et al. distinguished between two hypothetical frameworks for how apical dendrites might associate with partner glomeruli—the proximity model and the specificity model. While their observations seemed to align with the proximity model, we acknowledge that subsequent findings, particularly those from the Imai group (Aihara et al., 2021; Fujimoto et al., 2023), suggest that neuronal activity significantly influences mitral cell remodeling.

Taking into account these recent advances, we propose that an activity-dependent approach provides a compelling explanation for dendritic remodeling. In response to these developments, we have revised the "Results and Discussion" section to include a re-evaluation of our findings within the context of these newer models.

Thank you once again for bringing this to our attention; it has allowed us to present a more nuanced discussion of our results.

Referee #2:

This study evaluates the role of CASMAPs protein family in the development and function of olfactory circuits, with a particular focus on CASMAP2. The authors developed different genetic models to knock out the protein of interest. The absence of CASMAP2 led to an impairment in both mating behavior and basic olfactory functions in the male cohort. At cellular level, *Casmap2* ^{-/-} animals exhibited abnormalities in the Olfactory Bulb (OB) anatomy. Specifically, MCs were not only altered in their morphology but presented impairments at the level of dendrite pruning, key process for the establishment of the olfactory circuitry. These alterations led to an overall decrease in the activity of the circuitry, explaining the anosmia and mating-related impairments. Additionally, the authors briefly expand on the other members of the CASMAP family. *Casmap1* ^{-/-} animals showed defect in the migration of MCs while *Casmap3* ^{-/-} animals did not show any impairment.

The study is interesting and well executed. The significance of this work resides in a complete characterization of the CASMAP2's role in supporting olfaction, including a behavioral, reproductive and cellular perspective. This study may provide new general insight on how relevant the microtubule network is to cognitive functions. Beside this, the *Casmap2* ^{-/-} line may be considered and used as a model to study olfactory dysfunction and the role that MCs may play in that. I only have a few points of improvement to suggest.

Response:

We greatly appreciate your insightful and constructive feedback, which contributes to the enhancement of our work.

Major points

1. The experimental work is strong and data appear consistent with the general message of the study. However, the quality of the text is generally rather poor and, at times, difficult to understand. Advice from professional support on English writing is highly advised.

Response:

Thank you for your feedback. Your critique motivates us to improve. We have revised the manuscript and edited the English by professionals.

2. In different sections, claims are rather strong (e.g. data demonstrate/prove/indicate). However, as long as data are consistent with the claim, they often are not directly proving it. This again may have to do with English. Few examples of this general tendency are below:

-Line 175-181: claim that no differences in position and number of mature OSNs and other populations in the MOE is not supported by quantification. What is shown are qualitative images of MOE in the supplementary materials. Please tone down the claim (e.g.: "no major change was evident") or, alternatively, provide quantification.

Response:

Thanks for your advice. As non-native English speakers, we apologize for any confusion caused by inaccurate word usage. In the revised version of our manuscript, we carefully reviewed and revised the words “demonstrate,” “prove,” and “show.” Additionally, we sought the help of professional English writers to ensure the highest quality of our work.

Regarding lines 175-181(previous version of manuscript), we have removed this data and text from this version of the manuscript to preserve the integrity of the story.

-Line 241-250: claim that *Casmap2* *-/-* animals do not show defects in neither migration nor neurogenesis. This claim is supported by quantification of Tbr2+ and PGP9.5+ neurons at both developmental and adult stages but this alone is insufficient to directly prove "migration/neurogenesis". Combination of different BrdU paradigms for neuronal birthdating, stem cell proliferation, survival, etc., are needed to make the strong claims made by the authors.

Response:

Thank you for your suggestion. We have added new experimental data in the revised manuscript. Please refer to Figure EV3 for the details. In this figure, we performed double staining for BrdU and Pax6, which are neural progenitor markers, to analyze the neurogenesis of both WT and *Casmap2* *-/-* mice (see Figure EV3A and B). Additionally, we also conducted immunostaining of the olfactory bulb for Cleaved-Caspase 3 (Activated-Caspase 3) to analyze the apoptosis of both WT and *Casmap2* *-/-* mice (see Figure EV3C and D).

-Other instances of strong claims are present. Authors are advised to revised the whole manuscript.

Response:

Thanks for the suggestion. We have reviewed the text of this manuscript version and replaced any claims that lack supporting data or quantification.

3. Across the whole study the authors focused their interest on male animals, considering their infertility and impaired mating behavior. This is absolutely fine, but it would be potentially interesting to also evaluate how and if olfactory dysfunction and cellular alterations apply to female as well. Admittedly, this is very likely, and perhaps assessment may not need to be that thorough as it was in males. For example, simple pumps recognitions by smell and/or predator avoidance would be very simple to execute.

Response:

We appreciate your suggestion. During the revision period, we conducted the predator avoidance experiment and found that WT female mice can sense danger as they exhibit risk assessment behavior in the presence of a rat in the chamber. However, the number of risk assessment episodes in WT female mice was significantly lower in comparison to WT male mice. (WT male mice had a mean episode number of 47.4, while WT female mice had a mean episode number of 10.6). Also, unlike male mice,

WT female mice did not show significant avoidance behavior when they detected the scent of a predator. We believe that two possible reasons may account for these results. Firstly, male and female mice might have different evaluation systems, where the scent of a male rat could be defined as a dangerous signal by male mice while female mice may not. Secondly, the behavior of female mice could be influenced by their estrous cycle. Unfortunately, we cannot determine the possible reason without investing a significant amount of time into this question.

Although female WT and female *Camsap2* KO mice do not show differences in their avoidance behavior, we suspect that the absence of CAMSAP2 impairs female mice's sense of smell. We have two reasons to support this claim. Firstly, while WT female mice demonstrate risk assessment behavior, *Camsap2* KO female mice do not exhibit this behavior (Response letter Figure 1C, see below). Secondly, the dendrite morphology is abnormal in *Camsap2* KO female mice, as we did not differentiate between genders in the AAV-based experiments (Figure 3E-J and Figure 4C-H in the revised manuscript). However, although we have provided some possible explanations for this experiment, we cannot provide any solid data to prove our hypothesis. For this reason, we have decided not to include these results in our revised version of the manuscript.

Figure for referee with unpublished data and its description has been removed upon request by the authors.

Minor points

4. The last result section refers to KO models of CASMAP1 and CASMAP3 (line 314-332). To the reader, these results appear to be beyond the scope of this paper. The authors may consider to move them to the supplementary section, considering that the

strongest point of this study remains the characterization of *Casmap2*^{-/-}.

Response:

Thank you very much for your suggestion. We removed the results of CAMSAP1 and CAMSAP3 to focus on characterizing *Casmap2*^{-/-}.

5. Neither in the results section nor methods it is clearly stated how the histological quantification have been performed. Was the analysis performed stereologically collecting serial sections? Was it performed manually or automatically? Please state analysis details in the methods more clearly.

Response:

Thanks for your suggestion. We have added the detailed analysis in the methods section (Lines 437-448, Statistical analyses).

6. In different graphs (e. g. Fig. 1G, 2C, 2H) data points are poorly represented. Taking 1G as an example, the authors show that *Casmap2*^{-/-} male animals do not mate over the course of 5 days. In the graph, the data points for group 2 are all indicated at Day 5 as if all mice mated on this very day. However, this is not the case. Rather, the experiment was interrupted at day 5 and these mice never mated at all (this is my understanding). Indicating points at Day 5 is misleading, rather there should be no point at all, or an interrupted bar represented and/or the y axis changes accordingly.

Response:

We would like to express our gratitude to the reviewer for bringing to our attention a problem with the presentation of data in Figures 1G, 2C, and 2H in the previous version of our manuscript. We acknowledge that the data in those figures may have conveyed confusing messages. In this updated version of the manuscript, we have made some changes to address this issue. Specifically, we have used the number 0 to represent the mice that do not produce plugs in Figures 1G and 2H. Additionally, for Figure 2C, we have used the value of 1,000 seconds to denote the mice that failed to find food during the experiment. We hope that these modifications clarify the data and improve the overall quality of our manuscript.

Referee #3:

In this extremely ambitious and wide-ranging manuscript, the authors present a series of experiments investigating the role of CAMSAP genes in the development of the mouse olfactory system. They describe a number of phenotypes resulting from knockout of CAMSAP2 or CAMSAP1, some of which are convincingly supported by the data, but many of which are not. In particular, the conclusions drawn regarding causality of effects in these mouse models are problematic. It is clear that CAMSAP2 KO males are infertile, and that they have olfactory deficits, but it is not clear that the olfactory deficits are the cause of the infertility. More tenuous still is the proposed causal link between mitral cell dendritic abnormalities and olfactory deficits - it is absolutely not convincingly demonstrated in this paper that abnormal mitral cell dendritic development produces anosmia.

Without doing a huge amount of additional experimental work, it is not clear to me how these causal inferences could be sufficiently supported. Perhaps the most appropriate course of action would be to produce a much toned-down, descriptive version of the study which removes all of the currently uninterpretable data (current figures 3, 5I, 6 & 7), and which presents the reasonably solid findings that CAMSAP2 KO mice are infertile, that they have deficits in olfactory behaviour, and that they have abnormal mitral cell dendritic pruning, *without* attempting to draw causal links between these observations. This, in my view, could still form a useful addition to our understanding of the molecular processes underlying olfactory system development.

Response:

We appreciate your criticism and suggestions. While we believe our proposal regarding the causal link between infertile olfactory behavior and abnormal mitral cell dendrite morphogenesis is credible, we acknowledge that we cannot provide solid enough data to directly certify our conclusion due to the leakiness of AAV in Figure 6 of the former version of the manuscript. We cannot totally exclude the possibility that the leakiness of AAV may disturb the development of another region of the brain, as mentioned in our previous version of the manuscript. If we were to try to prove our conclusion, we would need to construct another mitral cell-specific Cre tool mouse line to specifically knockout the expression of *Camsap2* in the mitral cell. However, this work alone would take years to finish.

After a thorough discussion with our collaborators, we have concluded that the suggestion provided by Reviewer #3 is the wisest course for us. In this revised version of the manuscript, we have removed Figures 3, 5I, 6, and 7 and re-written the manuscript. The data in Figures 3, 6, and 7 will be revised with new experiments to support our proposal and published in the future.

Major issues

Link between infertility and olfactory deficits in the CAMSAP2 KO (Abstract line 32: 'knockout (KO) males were infertile due to impaired sense of smell'). The authors have done a thorough job of trying to identify potential sources of infertility and to narrow this down to a deficit in male copulatory behaviour. However, there is no

evidence that the lack of mounting is due to an inability to detect and/or respond to olfactory cues. It is entirely possible that the CAMSAP2 KO produces this effect via deficits in other, non-olfactory circuits involved in reproductive behaviour. This is especially likely given that, despite descriptions to the contrary, there *are* motor deficits on the rotorod test, together with a strong trend (with very small sample sizes presumably responsible for non-significance) towards less ultrasonic vocalisations in the KO (FigS2).

Response:

Thank you for your valuable suggestions. We have encountered similar queries during our research and have conducted multiple experiments to resolve this issue. Our study initially found that male mice with CAMSAP2 deficiency exhibited impaired mating behavior. Upon further investigation, we found that *Camsap2* KO mice were unable to perform pre-mating activities. As the olfactory and pheromonal signals are critical for the copulatory behavior (Chen & Hong, 2018), we suggest that the lack of olfaction in *Camsap2* KO mice results in a deficiency in male copulatory behavior.

We have two reasons to support our conclusion. Firstly, *Camsap2* KO mice have a defect in olfaction, which prevents them from accurately distinguishing olfactory cues, including the hormones released by females (as shown in Figure 2). Therefore, regardless of whether the olfactory circuit and non-olfactory circuits involved in reproductive behavior are functioning correctly or not, the inability to distinguish male and female caused by the defect in the olfactory bulb remains unchanged. It is analogous to the situation where we cannot collect water if the tap is closed, regardless of whether the pipes downstream of the tap are intact or not. Secondly, we tried to specifically knock out the expression of *Camsap2* in mitral cells, and these mice exhibited olfactory and reproductive behavioral defects (as shown in Figure 6 of the previous version of the manuscript). Therefore, we believe that the lack of mounting behavior is due to the olfactory defect.

In Figures 1H and I of the revised manuscript, it can be seen that the male mice with *Camsap2* KO show an almost complete absence of mounting behavior. However, in Figure EV1E, the motor ability of both WT and *Camsap2* KO male mice is comparable. During the revision process, we reanalyzed the data using two-way R-M ANOVA and found that *Camsap2* KO mice do not exhibit any motor deficits on the rotorod test. Therefore, we do not believe that the complete absence of mounting behavior is caused by any motor ability issues.

In Figures EV1A and EV1B, mice that lack the *Camsap2* gene (*Camsap2* KO mice) can still produce ultrasound. This result indicates that the vocal system of *Camsap2* KO mice could work properly. Therefore, the total loss of ultrasonic vocalizations in these mice is not due to an inability to engage in mating behavior but rather to a lack of motivation (Figure. 1J and K).

-It is important here to note that the purported 'mitral-cell-specific' CAMPSAP2 KO (Fig 6), which would potentially link olfaction and sterility, has actually not been shown to be MC-specific at all. Injecting AAV anywhere into the developing embryonic brain would be expected to produce widespread infection far from the injection site, and the authors show no evidence that M/TCS are the only cells in the nervous system affected by this manipulation. On the contrary, in fact, the labelling shown in Fig6E not only reveals off-target expression in OB granule cells (as

acknowledged by the authors) but also in other upstream areas of the OB such as the glomerular layer.

Response:

Your valuable suggestions are greatly appreciated. In Figure 6E (previous version of the manuscript), as we mentioned, we cannot completely exclude the possibility of off-target expression. What we can prove is that the mitral cell is the main neuron type which has been manipulated by AAV. We agree that the conclusion “MC-specific deficiency of CAMSAP2 alters odor-induced behaviors” was hastily taken and unsolid enough to be criticized. In this revised manuscript, we have removed Figure 6 and associated conclusions.

- Link between mitral cell pruning and olfactory deficits in the CAMSAP2 KO. Strong claims are made about the underlying causes of olfactory deficits in this KO, e.g. Abstract line 33: 'The dysosmia was due to the pruning defect of MC and the misconnection between OSNs and MCs.' In fact, severe olfactory deficits being produced specifically by changes in mitral cell dendritic development would be an extremely surprising finding, given the wild-type accessory OB (AOB) functions extremely well with mitral cells that target multiple glomeruli. Moreover, hugely disrupted OB circuitry, including major mitral cell dendritic abnormalities, has been shown to be associated with strikingly few overt olfactory deficits (see recent publication from the Rokni lab). Here it is clear, novel, and interesting that MC dendritic pruning is affected by (cell-autonomous) CAMPSAP2 KO - as stated above, I think this is useful information for the field of olfactory development. However, there is no clear evidence in this manuscript that this pruning defect underlies the severe olfactory deficits observed. See above for issues with the 'MC-specific' manipulation that is the major piece of evidence used to make this claim - the authors simply cannot claim that 'our data indicate that MC-specific deficiency of CAMSAP2 is the primary cause of dysosmia.' (line 313).

Response:

Thank you for your constructive feedback. We have read the work from the Rokni lab and find it to be a very interesting story. In their research, the olfactory bulbs have undergone near-total loss and some olfactory sensory neuron axons have ectopically projected to the olfactory cortical region. However, in our mice model, the olfactory sensory neuron axons only project to the glomeruli of the olfactory bulb (as shown in Figure S6C in the previous version of our manuscript). This difference in olfactory bulb structure may have led to the different results observed in our study. However, further experimental data is needed to support this proposal. We plan to explore this question further in our future studies using other suitable Cre-tool mice. In the meantime, we have altered our conclusions in the abstract and the manuscript in light of your valid criticisms.

- The authors focused on the development/pruning/projection of M/TC apical dendrites, but what about the lateral dendrites? Deficits here would be more closely related to the distribution of CAMSAP2 immunolabel shown in Fig5B, and would be far more likely than apical dendrite abnormalities to be related to the reported (convincing) reduction in EPL width.

Response:

Thanks for this suggestion. We focused our research on the “development/pruning/projection” of mitral cell apical dendrites as it is the most obvious phenotype we can find. With the reminder by reviewer#1 and reviewer#3, we think the lateral dendrites may also be affected. In this revised manuscript, we replaced the word “apical dendrite pruning” with “dendrite morphogenesis” and discussed the possibility of lateral dendrite effects in the text.

- Issues with activity marker experiments (Fig 3, plus supplementals). There are no non-stimulation controls for baseline marker expression, no co-stains to identify the labelled cell types, no attempt to standardise labelling/imaging/analysis across conditions, and at best the most rudimentary 'quantification' involving cell counts that is not described at all in the Methods. In the *Egr1* example, for instance, it seems that the staining is weaker in the cKO, though the authors claim otherwise. I think these experiments are completely uninterpretable, and should be removed from the paper.

Response:

Thanks for the constructive criticism. It is our mistake in this experiment. As we mentioned above, we have removed Figure 3 in this revised manuscript, and we will add the negative controls in our future works.

- Issues with the 'OSN-MC connectivity' experiment (Fig3FG). Here a single uncontrolled, highly unspecific AAV injection protocol is used to try to infer connectivity between OSNs and OB neurons. There is no demonstration that the resultant labelling in OB cells is indeed 'transsynaptic', rather than produced by a direct leak of AAV injected into (very physically close) olfactory epithelium. Even if the OB cell label here is (all) transsynaptic, there are many, many reasons why it might be different in the KO vs WT, especially when the gene involved controls microtubule function - the authors presumably assume, but cannot be sure, that AAV transport/synaptic transfer/postsynaptic uptake are all equal in the CAMSAP2 KO. This single, highly flawed approach cannot possibly be used to conclude a 'loss of synapses between OSNs and MCs'! Again, I strongly suggest that these data are removed from the paper.

Response:

Thanks for this suggestion.

In fact, the possibility of AAV leakage in our experimental system is very small. As 300 nL is not a large volume, it is not easily leaked from the olfactory epithelium into the olfactory bulb. If the AAV leak from the olfactory epithelium to the olfactory bulb, the overwhelming majority of labeled neurons, if not all, would be periglomerular neurons, and it would be difficult to label the mitral cells. Another experiment to inject the non-trans synapse AAV into olfactory epithelium would be the best solution to verify this conclusion.

Secondly, we agree with your view that AAV transport and synaptic connectivity are closely related. Our current data cannot exclude the possibility that deficiency of CAMSAP2 does not alter virus transport/synaptic transfer/postsynaptic uptake.

As mentioned above, we have removed Figure 3 in this manuscript but will keep your

points in mind in our future studies.

- Lack of quantification throughout the paper. This includes important statements made on the basis of single qualitative examples regarding OE organisation, non-EPL OB layers/glomerular characterisation, the MC-specific KO of CAMSAP2, MCL organisation at P0 and its 'obscure' borders later, CAMSAP2 immuno, all of the CAMSAP1/3 data...

Response:

Thanks for this suggestion. As mentioned above, we have removed the data of OE and Figure 3, Figure 6, Figure 7. In our next work, we will add these quantification data in figures.

- The data in Fig 5I address a completely different question unrelated to the rest of the manuscript and do not appear to complement the wider study. I suggest these are also removed. If they are retained (with a clear rationale), there is a missing input control for CAMSAP2-BirA*-HA alone, just like conditions 1 and 2. How do we know that whatever we see in the HA blot in 3 and 4 does correspond indeed to the CAMSAP2 band? Could it be a non-specific binding of the primary? Lanes 1 and 3 in input look strangely similar in the anti GFP blot. Also, lines 271 to 273, the authors claim that 'Our results indicated that CAMSAP2 strongly interacts with Katanin P60L1, while we only occasionally found a very weak interaction between CAMSAP2 and Kif2a'. While I agree with the first statement (and even so, I would expect the CAMSAP2 band to be stronger in the pull-down conditions compared to the input, as the pull-down per se should enrich the quantity of detected protein), looking at Figure 5I, the second part is not true: if we look at the lane 4 in the pull-down panel, there is a strong band for the HA blot (indicative of the presence of CAMSAP2), but there is nothing in the GFP immunoblot. It is therefore impossible to conclude that there is an interaction from this experiment. I suggest the authors to provide an image that supports their claims, or, alternatively, remove such claim from the main text.

Response:

Thanks for this suggestion. We have removed this figure panel and associated text in this revised version of the manuscript.

- Contrary to the stated conclusions (e.g. Abstract line 34), there is not a convincing MTC migratory phenotype in the CAMSAP1 KO. Instead, all that is shown is an unquantified example image of Tbr2+ cells in the GCL. How can the authors be sure that this is not just aberrant Tbr2 expression in granule cells (or other GCL cells)? Would e.g. Golgi staining show clear MTC morphology of GCL-localised cells in the CAMSAP1 KO?

Response:

Thanks for this suggestion. As mentioned above, we have removed Figure 7 in this revised manuscript. We agree that aberrant Tbr2 expression may cause this phenotype, immunostaining for another mitral cell marker (e.g. Tbr1, Tbx21, Pcdh21) would be a wise choice to verify our conclusion, and this experiment is on our schedule for future studies. Thanks again.

Minor issues:

- Did the controls for the 'MC-specific' KO manipulation comprise non-floxed mice that were also injected? This should have been the case. If not, all of the effects of this manipulation can probably be explained by non-specific effects of the injection procedure. But if so, why do the authors interpret their results in terms of possible dorsal OB damage (line 300)? Wouldn't similar damage have been caused by the injections in non-floxed control mice?

Response:

Thank you for your comments. We injected the non-floxed mice with AAV. In the previous manuscript, we described one possible explanation for this (Line 300 of the previous manuscript). We were also very puzzled when we got this result (Figure 6D). Our proposal is that this injection procedure may impair the dorsal region of OB, and this impairment may be much worse in KO mice, as the deficiency of CAMSAP2 alters the development of the mitral cell itself.

However, we do acknowledge that this is just one of the possible reasons. In this manuscript, we have removed these data.

Recent work from the Imai group has significantly advanced our understanding of the molecular mechanisms of mitral cell dendritic pruning. Why isn't this important and highly relevant work cited?

Response:

When we were drafting the manuscript, the work done by the Imai group had not been published, which is why we did not reference it. However, reviewer #1 and reviewer #3 reminded us about this work, after which we read the paper and found it to be very interesting and influential. We have now cited this paper in our manuscript and discussed our new insights based on their findings.

- All statistical tests are simple pairwise t-tests. Were assumptions for parametric tests met? Repeated tests on the same data will increase the rate of false positive 'significant' findings, so the following data (if included in any revision) would need reanalysing with appropriate tests that take multiple comparisons into consideration: Fig1G (needs 1-way ANOVA); Fig2A,B,F,J and 6D (I recommend taking ratios for each mouse, then comparing those with a t-test); Fig5F (needs 2-way ANOVA); FigS1E (needs 2-way R-M ANOVA); FigS2E (also needs 2-way R-M ANOVA); FigS6F (needs 2-way R-M ANOVA, or some kind of integrated measure per animal)

Response:

Thanks for this suggestion. In our previous manuscript, unpaired t-tests were performed to analyze the data as we assume these results fit the normal distribution. We are using the Student's t-test to analyze our data, as it is a widely accepted statistical method (Liu *et al*, 2017; Matsuo *et al*, 2015; Minegishi *et al*, 2018; Ung *et al*, 2021).

During the revision period, we reanalyzed the data in this manuscript as suggested by the reviewer, and most of the data acquired similar results. We also found some

differences between unpaired t-tests and two-way R-M ANOVA; one is the body weight of female mice, and the other is the motor performance on the rotarod (Response letter Figure 2, see below). However, these differences do not affect our conclusion in this study. Some even support our conclusion more powerfully (Figure EV1E, the motor ability is not responsible for the mating behavioral defect of *Camsap2* KO mice).

We feel badly that we labeled the wrong sample size of MCs in Figure 3D (Figure 4D in the previous manuscript). To avoid these mistakes, we rechecked our original data and found another 4 mislabeling sample sizes (Figure 4F, Figure EV1E, Figure EV2 and Appendix Figure S1E in this manuscript). The data we displayed in the figure are right, while we typed the wrong number when we organized these pictures in Photoshop. we have corrected these errors in this manuscript.

In the end, we thank the reviewer again for providing such professional suggestions on the statistical methods.

Figure for referee with unpublished data and its description has been removed upon request by the authors.

As we made revisions, some figures were removed, resulting in changes to the numbers in the following table.

Figures in previous manuscript	Figures in revised manuscript
Figure 4D	Figure 3D
Figure 5F	Figure 4F
Figure S1E	Appendix Figure S1E
Figure S2E	Figure EV1E
Figure S6D-H	Figure EV2A-E

- I do not understand the reasoning on Line 628, which appears to argue that statistical tests were not performed on proportion data because they gave non-significant results (!). Many tests are available for specifically assessing differences in proportions, and these actually tend to be rather powerful.

Response:

The sample size in one section is small (sometimes, we can only see very few neurons in one section), and we worried that this sample size may give some false positive results. In this revised manuscript, we defined the sections from one mouse as one sample to analyze the percentage of neuronal type, and statistical tests were performed on these data.

- The Methods section needs a lot more detail on how images were acquired, and how they were analysed. Were experimenters blind to genotype/manipulation in this study? If not, how can the authors discount implicit bias in all of their data, especially the behavioural tests?

Response:

The experiment was blinded to the experimenters during the behavioral assay. In this revised manuscript, we have rewritten the methods and added more experimental detail.

- Scalebars are missing from many of the images.

Response:

Thanks for catching the mistake. We have added the missing scale bars to the manuscript.

References

- Aihara S, Fujimoto S, Sakaguchi R, Imai T (2021) BMPR-2 gates activity-dependent stabilization of primary dendrites during mitral cell remodeling. *Cell Rep* 35: 109276
- Chen P, Hong W (2018) Neural Circuit Mechanisms of Social Behavior. *Neuron* 98: 16-30
- Fujimoto S, Leiwe MN, Aihara S, Sakaguchi R, Muroyama Y, Kobayakawa R, Kobayakawa K, Saito T, Imai T (2023) Activity-dependent local protection and lateral inhibition control synaptic competition in developing mitral cells in mice. *Dev Cell* 58: 1221-1236 e1227
- Liu Z, Chen Z, Shang C, Yan F, Shi Y, Zhang J, Qu B, Han H, Wang Y, Li D *et al* (2017) IGF1-Dependent Synaptic Plasticity of Mitral Cells in Olfactory Memory during Social Learning. *Neuron* 95: 106-122 e105
- Matsuo T, Hattori T, Asaba A, Inoue N, Kanomata N, Kikusui T, Kobayakawa R, Kobayakawa K (2015) Genetic dissection of pheromone processing reveals main olfactory system-mediated social behaviors in mice. *Proc Natl Acad Sci U S A* 112: E311-320
- Minegishi T, Uesugi Y, Kaneko N, Yoshida W, Sawamoto K, Inagaki N (2018) Shootin1b Mediates a Mechanical Clutch to Produce Force for Neuronal Migration. *Cell Rep* 25: 624-639 e626
- Nagae S, Meng W, Takeichi M (2013) Non-centrosomal microtubules regulate F-actin organization through the suppression of GEF-H1 activity. *Genes Cells* 18: 387-396
- Nishizumi H, Miyashita A, Inoue N, Inokuchi K, Aoki M, Sakano H (2019) Primary dendrites of mitral cells synapse unto neighboring glomeruli independent of their odorant receptor identity. *Commun Biol* 2: 14
- Ung K, Huang TW, Lozzi B, Woo J, Hanson E, Pekarek B, Tepe B, Sardar D, Cheng YT, Liu G *et al* (2021) Olfactory bulb astrocytes mediate sensory circuit processing through Sox9 in the mouse brain. *Nat Commun* 12: 5230
- Zhou Z, Xu H, Li Y, Yang M, Zhang R, Shiraishi A, Kiyonari H, Liang X, Huang X, Wang Y *et al* (2020) CAMSAP1 breaks the homeostatic microtubule network to instruct neuronal polarity. *Proc Natl Acad Sci U S A* 117: 22193-22203

Dear Prof. Meng,

Thank you for submitting your revised manuscript. It has now been seen by all of the original referees.

As you can see, the referees find that the study is significantly improved during revision and recommend publication. However, I need you to address the points below before I can accept the manuscript.

- Please address the remaining minor concerns of referee #3.
- Please remove the Author Contributions section from the manuscript.
- We note the phrase 'Our previously unpublished data...' on page 6, which is not allowed as per EMBO Press policy. Therefore, please either show the data or remove the phrase.
- We note that the grant numbers for the funders are currently missing from the manuscript submission system - i.e. 32100760, 31930025, and 32270736; 2018YFA0801104 and 2021YFA0804802; and 510858072.
- We note the following incorrect figure callouts in the manuscript: "Appendix Fig EV1A and B", "Appendix Fig EV1C and D", "Appendix Fig EV1E", which should be either "Appendix Fig S#" or "Fig EV#".
- Please resubmit Source Data as one file per figure. Source data files need to be submitted as zipped folders, one .zip file for each figure. Inside each folder, the files should be organized in subfolders, one subfolder for each panel.
- Please include the statement "For most microscopic images, we provided two type figures, the edited JPG files (easy reading), and the original high resolution tiff files." in the source data checklist. You can use the blue comment box for it.
- Our production/data editors have asked you to clarify several points in the figure legends:
 - o Please note that a separate 'Data Information' entitled section is required in the legends of figures 1a-i, k; 2a-c, e-j; 3a-d, f, h, j; 4b-d, f-h; EV 1b-e; EV 2b-e; EV 3a-d; EV 4a-e.
 - o Please note that in figures EV 1b-d; there is a mismatch between the annotated p values in the figure legend and the annotated p values in the figure file that should be corrected.

Thank you again for giving us to consider your manuscript for EMBO Reports, I look forward to your minor revision.

Kind regards,

Deniz Senyilmaz Tiebe

--

Deniz Senyilmaz Tiebe, PhD
Editor
EMBO Reports

Referee #1:

The authors have adequately responded to our previous comments.
One minor comment. In line 53, the connection between the ORs and the olfactory cortex is not topographic.

Referee #2:

this is an interesting study. in their revision, the authors have satisfactorily addressed all my pending points and suggestions.

Referee #3:

This thoroughly revised manuscript addresses all of my major concerns. I just have a few remaining suggestions to improve clarity:

- The authors have been commendably careful with their claims about causality in this revised version. There are just a couple of places where this could be improved. First, on line 143-4 'As Camsap2(-/-) males sniff but do not mate with the receptive females (Fig 1G-I), we then asked whether it is an olfaction defect that causes these results'. It would be better to steer away from the suggestion that olfactory deficits definitely cause infertility by saying 'might cause these results' instead. Second, on line 251 the statement is a little too strong. 'All of these defects will disrupt the wiring of the olfactory circuit and impair the detection or discrimination of chemical cues', should instead read 'All of these defects would be predicted to disrupt' or similar. It would also help to clarify interpretations of the data if there were an explicit statement in the Discussion section which says that defects in MTC pruning may contribute to the olfactory deficits observed in CAMSAP2 KOs, but that other (as yet unassessed) differences in the KO olfactory system may also play an important role.
- The authors appear to have now employed appropriate statistical tests for their datasets. For the 2-way ANOVAs, they need to

detail the full results of these tests, including both main effects, the interaction effect, and the results of any post-hoc tests performed. For example, for the data in 4F, 'effect of age, $p = ?$; effect of genotype, $p = ?$, effect of age x genotype interaction, $p = ?$; Tukey post-test (or whichever was employed) $+/+$ vs fl/fl at P0, $p = ?$; $+/+$ vs fl/fl at P7, $p = ?$).' This will allow readers to understand how exactly the data were analysed. In general, it is important to report the full p value for each test, rather than just summaries as 'n.s. vs $^{**}/^{***}$ '.

Response:

- **Please address the remaining minor concerns of referee #3.**

Response:

Thank you very much for your suggestion. We have addressed referee #3's minor concerns. Please refer to our response to referee #3 for more details.

- **Please remove the Author Contributions section from the manuscript.**

Response:

We have removed the Author Contributions section from the manuscript.

- **We note the phrase 'Our previously unpublished data...' on page 6, which is not allowed as per EMBO Press policy. Therefore, please either show the data or remove the phrase.**

Response:

We have replaced the phrase "Our previously unpublished data..." with "As the OB is an important station for odor information processing and delivery" on page 6 (lines 177-178 of the previous manuscript, and line 176 in this manuscript).

- **We note that the grant numbers for the funders are currently missing from the manuscript submission system - i.e. 32100760, 31930025, and 32270736; 2018YFA0801104 and 2021YFA0804802; and 510858072.**

Response:

We have now included the grant numbers of the funders in the manuscript submission system.

- **We note the following incorrect figure callouts in the manuscript: "Appendix Fig EV1A and B", "Appendix Fig EV1C and D", "Appendix Fig EV1E", which should be either "Appendix Fig S#" or "Fig EV#".**

Response:

We have corrected the figure callouts in the manuscript: "Appendix Fig EV1A and B", "Appendix Fig EV1C and D", and "Appendix Fig EV1E" (Lines 131, 137, and 139).

- **Please resubmit Source Data as one file per figure. Source data files need to be submitted as zipped folders, one .zip file for each figure. Inside each folder, the files should be organized in subfolders, one subfolder for each panel.**

Response:

We have resubmitted the Source Data following the guidance.

- **Please include the statement "For most microscopic images, we provided two type figures, the edited JPG files (easy reading), and the original high resolution tiff files." in the source data checklist. You can use the blue comment box for it.**

Response:

We have included the statement "For most microscopic images, we provided two type figures, the edited JPG files (easy reading), and the original high resolution tiff files." in the blue comment box of source data checklist.

- **Our production/data editors have asked you to clarify several points in the figure legends:**
 - o **Please note that a separate 'Data Information' entitled section is required in the legends of figures 1a-i, k; 2a-c, e-j; 3a-d, f, h, j; 4b-d, f-h; EV 1b-e; EV 2b-e; EV 3a-d; EV 4a-e.**

Response:

We have added the separate "Data Information" entitled section in the new figure legends.

- **Please note that in figures EV 1b-d; there is a mismatch between the annotated p values in the figure legend and the annotated p values in the figure file that should be corrected.**

Response:

We have corrected the mismatch between the annotated p values in the figure legend and figures EV1 b-d. Besides, we also corrected the mismatch of annotated p values between figure 2G and its figure legend.

Referee #1:

- **The authors have adequately responded to our previous comments.**

One minor comment. In line 53, the connection between the ORs and the olfactory cortex is not topographic.

Response:

Many thanks for correcting our mistake. We have replaced the word "topographical" with "precise" in line 53.

Referee #2:

- **This is an interesting study. In their revision, the authors have satisfactorily addressed all my pending points and suggestions.**

Response:

We appreciate your interest in our research. Thank you for your feedback and contributions.

Referee #3:

This thoroughly revised manuscript addresses all of my major concerns. I just have a few remaining suggestions to improve clarity:

- **The authors have been commendably careful with their claims about causality in this revised version. There are just a couple of places where this could be improved. First, on line 143-4 'As Camsap2(-/-) males sniff but do not mate with the receptive females (Fig 1G-I), we then asked whether it is an olfaction defect that causes these results'. It would be better to steer away from the suggestion that olfactory deficits definitely cause infertility by saying 'might cause these results' instead. Second, on line 251 the statement is a little too strong. 'All of these defects will disrupt the wiring of the olfactory circuit and impair the detection or discrimination of chemical cues', should instead read 'All of these defects would be predicted to disrupt' or similar. It would also help to clarify interpretations of the data if there were an explicit statement in the Discussion section which says that defects in MTC pruning may contribute to the olfactory deficits observed in CAMSAP2 KOs, but that other (as yet unassessed) differences in the KO olfactory system may also play an important role.**

Response:

We would like to thank you for your interest in our study. Your contribution is greatly appreciated, and we are pleased to address your concern regarding.

As suggestion, we have added the word “might” in line 143. In line 250, we have replaced the words “will disrupt” with “would be predicted to disrupt” to weaken our statement. In lines 258-260, we have added an explicit statement as recommendation.

• The authors appear to have now employed appropriate statistical tests for their datasets. For the 2-way ANOVAs, they need to detail the full results of these tests, including both main effects, the interaction effect, and the results of any post-hoc tests performed. For example, for the data in 4F, 'effect of age, $p = ?$; effect of genotype, $p = ?$, effect of age x genotype interaction, $p = ?$; Tukey post-test (or whichever was employed) $+/+$ vs fl/fl at P0, $p = ?$; $+/+$ vs fl/fl at P7, $p = ?$).' This will allow readers to understand how exactly the data were analysed. In general, it is important to report the full p value for each test, rather than just summaries as 'n.s. vs $*//**$ '.**

Response:

Response:

We have added much more detail information of statistical tests in figure legends, we also report the full p-value for each test in figure legends.

Dear Wenxiang,

Thank you for submitting your revised manuscript. I have now looked at everything and all is fine. Therefore, I am very pleased to accept your manuscript for publication in EMBO Reports.

Congratulations on a nice work!

Kind regards,

Deniz

--

Deniz Senyilmaz Tiebe, PhD

Editor

EMBO Reports

--
